



# Comprehensive Global Assessment of 23 Gridded Precipitation Datasets Across 16,295 Catchments Using Hydrological Modeling

Ather Abbas[1], Yuan Yang[2], Ming Pan[2], Yves Tramblay[3], Chaopeng Shen[4], Haoyu Ji[4], Solomon H. Gebrechorkos[5], Florian Pappenberger[6], JongCheol Pyo[7], Dapeng Feng[8], George Huffman[9], Phu Nguyen[10], Christian Massari[11], Luca Brocca[11], Tan Jackson[9], and Hylke E. Beck[1]

[1]Division of Physical Science and Engineering, King Abdullah University of Science and Technology, Thuwal 23955-6900, Saudi Arabia
[2]Center for Western Weather and Water Extremes, Scripps Institution of Oceanography, University of California San Diego, La Jolla, CA, USA
[3]Espace Dev (University Montpellier, IRD), Montpellier, France
[4]Civil and Environmental Engineering, The Pennsylvania State University, PA, USA
[5]School of Geography and the Environment, University of Oxford, Oxford, UK
[6]European Centre for Medium-range Weather Forecasts, Reading/Bonn/Bologna, UK/DE/IT
[7]Department of Environmental Engineering, Pusan National University, Busan, 46241, Republic of Korea
[8]Department of Earth System Science, Stanford University, Stanford, CA 94305, USA
[9]NASA Goddard Space Flight Center, Greenbelt, MD, USA
[10]Center for Hydrometeorology and Remote Sensing (CHRS), Department of Civil and Environmental Engineering, University of California, Irvine, CA 92697, USA
[11]Research Institute for Geo-Hydrological Protection (CNR-IRPI), National Research Council, Perugia, Italy

**Correspondence:** Hylke E. Beck (hylke.beck@kaust.edu.sa)

**Abstract.** Numerous gridded precipitation ($P$) datasets have been developed to address a variety of needs and challenges. However, selecting the most suitable and reliable dataset remains a challenge for users. We conducted the most comprehensive global evaluation to date of gridded (sub-)daily $P$ datasets using hydrological modeling. A total of 23 datasets, derived from satellite, model, gauge sources, or their combinations thereof, were assessed. To evaluate their performance, we calibrated the

conceptual hydrological model HBV against observed daily streamflow for 16,295 catchments (each $< 10,000$ km$^2$) worldwide, using each $P$ dataset as input. The Kling-Gupta Efficiency (KGE) was used as the performance metric and the calibration score served as a proxy for $P$ dataset performance. Overall, MSWEP V2.8 demonstrated the highest performance (median KGE of 0.75), highlighting the value of merging $P$ estimates from diverse data sources and applying daily gauge corrections. Among the purely satellite-based $P$ datasets, the soil moisture- and microwave-based GPM+SM2RAIN dataset

performed best (median KGE of 0.60), while the JRA-3Q reanalysis ranked highest among the purely model-based datasets (median KGE of 0.67), outperforming the widely used ERA5 reanalysis (median KGE of 0.59). Performance varied across Köppen-Geiger climate zones, with the best results in polar (E) regions (median KGE of 0.74 across datasets) and the lowest in arid (B) regions (median KGE of 0.33 across datasets). We further examined the spatial relationships between catchment attributes and KGE scores, identifying potential evaporation, air temperature, solid $P$ fraction, and latitude as the strongest

predictors of performance. Our analysis revealed significant regional differences in dataset performance and heterogeneity in





$P$ error characteristics, underscoring the critical importance of careful dataset selection for water resource management, hazard assessment, agricultural planning, and environmental monitoring.

## 1 Introduction

Understanding the spatio-temporal distribution of precipitation ($P$) is crucial for a wide range of applications, including water
resources assessment, flood forecasting, agricultural monitoring, and disease tracking (Liang and Gornish, 2019; Hinge et al., 2022; Dimitrova et al., 2022). However, $P$ exhibits high variability across space and time, making it difficult to estimate, particularly in regions with complex topography, convection-driven precipitation, or snow-dominated climates (Herold et al., 2016; Prein and Gobiet, 2017; Sharma et al., 2020; Li et al., 2020; Tarek et al., 2021). $P$ estimates can be derived from satellites, models, and rain gauges, but each data source is subject to limitations. Satellite retrievals are affected by surface
snow and ice (Cao et al., 2018; Chen et al., 2020), and snowfall detection remains challenging (You et al., 2021; Jääskeläinen et al., 2024; Girotto et al., 2024b). Reanalyses (e.g., ERA5, Hersbach et al., 2020) rely on uncertain parameterizations and often lack sufficient spatial resolution to adequately capture orographic effects (Skamarock, 2004; Ménégoz et al., 2013; Liu et al., 2018). Rain gauge networks, are sparse and biased towards lower elevations (Schneider et al., 2014; Kidd et al., 2017; Ehsani and Behrangi, 2022), and gauges can severely underestimate snowfall due to wind-induced under-catch (Groisman and
Legates, 1994; Sevruk et al., 2009; Rasmussen et al., 2012; Girotto et al., 2024a).

In recent decades, numerous gridded $P$ datasets have been developed based on these data sources and combinations thereof. Each dataset has a different design objectives, spatio-temporal resolution, coverage, algorithms, and latency (see Table 1 for an overview of quasi- and fully-global datasets). A plethora of studies have evaluated these datasets (see, e.g., reviews by Gebremichael, 2010, Maggioni et al., 2016, and Sun et al., 2018). However, the large majority of these studies use rain gauge
observations as reference, which has limitations: (i) rain gauge observations are unavailable in many regions (Kidd et al., 2017); (ii) differences in scale between point-based rain gauges and grid-based $P$ datasets (Ensor and Robeson, 2008; Yates et al., 2006) can skew results; (iii) time discrepancies between daily accumulations of gauges and satellite and (re)analysis datasets (Yang et al., 2020; Beck et al., 2019b) can yield misleading daily evaluation results; (iv) the systematic $P$ underestimation by rain gauges in snow-dominated and mountainous regions (Groisman and Legates, 1994; Sevruk et al., 2009; Rasmussen et al.,
2012) can unfairly penalize $P$ datasets in these regions; and (iv) using rain gauges already incorporated into the $P$ datasets for validation results in misleading conclusions.

An alternative approach to evaluate $P$ datasets is to use hydrological modeling, wherein streamflow simulations driven by different $P$ datasets are compared to observed streamflow. The degree of correspondence between simulated and observed streamflow serves as a proxy for how accurately the $P$ dataset captures the intensity and timing of $P$ events. This approach
avoids the aforementioned limitations by providing a direct, real-world measure of performance that reflects the dataset's ability to capture $P$ dynamics in a hydrological context (Camici et al., 2018). Several studies have successfully employed this approach to evaluate various $P$ datasets (e.g., Voisin et al., 2008; Su et al., 2008; Bitew et al., 2012; Tang et al., 2016; Beck et al., 2017b; Lussana et al., 2018; Mazzoleni et al., 2019; Pradhan and Indu, 2021; Xiang et al., 2021; Gu et al., 2023; Gebrechorkos et al.,



2023). However, many studies are limited in scope by (i) focusing on specific regions or subcontinents, or using streamflow
data from relatively few catchments, thus restricting the generalizability of their findings; (ii) analyzing only a small subset
of available $P$ datasets, often excluding model-based datasets; (iii) focusing on a monthly rather than daily time scale, which
can obscure important short-term variability, such as extreme rainfall events or floods. Additionally, several studies failed to
re-calibrate the hydrological model for each $P$ dataset, including the recent global assessment by Gebrechorkos et al. (2023),
which could result n biased conclusions.

Here we present the most comprehensive evaluation of gridded (sub-)daily (quasi-)global $P$ datasets to date. We leverage an
unparalleled database of streamflow observations from 16,295 catchments worldwide, spanning all climate zones and latitudes,
to ensure broad generalizability of our results. Moreover, we evaluate an extensive collection of 23 $P$ datasets, including new
datasets like the microwave-based IMERG V7 (Huffman et al., 2019), the infrared-based PDIR-Now (Nguyen et al., 2020),
and the reanalysis JRA-3Q (Kosaka et al., 2024), all three of which have not been comprehensively assessed at the global scale
yet. To provide a fair and balanced assessment, we re-calibrate the hydrological model for each $P$ dataset.

## 2 Data and Methods

### 2.1 Gridded $P$ Datasets

Table 1 lists the 23 gridded $P$ datasets used in this evaluation. These datasets were selected based on their global or quasi-
global coverage, widespread use in hydrological applications, and availability of daily or sub-daily data. Regional datasets,
while valuable, were excluded to maintain consistency across diverse geographic areas (e.g., APHRODITE for Asia, NLDAS
for North America). These datasets are tailored for specific purposes: some, like IMERG-E and PDIR-Now, are designed for
short-latency applications such as monitoring heavy $P$ events, while others with longer latency, such as CHIRPS V2.0 and
IMERG-Final V7, are more suitable for comprehensive, long-term climate and hydrological analyses.

The datasets fall into two main categories: non-gauge-based, which rely entirely on satellite and/or model data for their tem-
poral dynamics, and gauge-based, which rely at least partially on rain gauge observations for their temporal dynamics (thereby
precluding an independent evaluation with rain gauge data). We included 12 datasets in our evaluation that are solely based on
satellite data (IMERG-Early V6, IMERG-Late V6 and V7, PERSIANN-CCS, PDIR-Now, GSMaP-std V7 and V8, SM2RAIN-
ASCAT and -CCI, GPM+SM2RAIN, and CMORPH-RAW and -RT), three that are exclusively model-based (ERA5, GDAS,
and JRA-3Q), and one based only on rain gauge data (CPC Unified). Additionally, we included three datasets that combine
both satellite and gauge observations (GPCP V3.2 and IMERG-Final V7, PERSIANN-CCS-CDR), as well as two that combine
satellite, model, and gauge data (CHIRPS V2.0 and MSWEP V2.8), and two that combine model and satellite data (CHIRP
and MSWEP-ng V2.8). For transparency and reproducibility, we explicitly indicate the version numbers throughout the text
for all datasets with available version information.

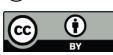

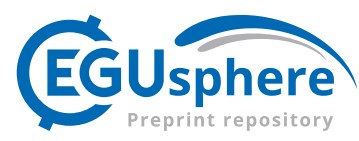

**Table 1.** Overview of the (sub-)daily (quasi-)global gridded *P* datasets evaluated in this study. Definition of abbreviations: S=satellite, G=gauge, M=Model.

| Data | Data Source | Temporal Res. | Spatial. Res. | Spatial Cov. | Temp. Cov. | Time Latency | Reference |
|---|---|---|---|---|---|---|---|
| CHIRP | S,M | Daily | 0.05° | Land, 50° N/S | 1981–NRT | 6 days | Funk et al. (2015) |
| CHIRPS V2.0 | S,G,M | Daily | 0.05° | Land, 50° N/S | 1981–NRT | 2 weeks | Funk et al. (2015) |
| CMORPH-RAW | S | 30 min | 8 km | 60° N/S | 2019 - NRT | 4 hours | Joyce et al. (2004) |
| CMORPH-RT | S | 30 min | 8 km | 60° N/S | 2019 - NRT | 4 hours | Xie et al. (2017) |
| CPC Unified | G | Daily | 0.5° | Land | 1979–NRT | 1 day | Chen et al. (2008) |
| ERA5 | M | Hourly | 0.25° | Global | 1940–NRT | 6 days | Hersbach et al. (2020) |
| GDAS | M | Hourly | 0.25° | Global | 2001–NRT | 3-6 hours | NCEP (2024) |
| GPCP V3.2 | S, G | daily | 0.5° | Global | 2000 - NRT |  | Huffman et al. (2023) |
| IMERG-Final V7 | S, G | 30 min. | 0.1° | Global | 2000–NRT | 3 months | Huffman et al. (2019) |
| IMERG-Late V7 | S | 30 min. | 0.1° | Global | 2000–NRT | 12 hours | Huffman et al. (2019) |
| IMERG-Late V6 | S | 30 min. | 0.1° | 60° N/S | 2000–2024 | 12 hours | Huffman et al. (2019) |
| IMERG-Early V7 | S | 30 min | 0.1° | 60° N/S | 2000 - NRT | 4 hours | Huffman et al. (2019) |
| GSMaP-std V7 | S | Hourly | 0.1° | 60° N/S | 2000–NRT |  | Kubota et al. (2020) |
| GSMaP-std V8 | S | Hourly | 0.1° | 60° N/S | 2000–NRT |  | Kubota et al. (2024) |
| JRA-3Q | M | 3-hourly | ~40 km | Global | 1947–NRT | 20 days | Kosaka et al. (2024) |
| MSWEP V2.8 | S,G,M | 3-hourly | 0.1° | Global | 1979–NRT | 3 hours | Beck et al. (2019b) |
| MSWEP-ng V2.8 | S,M | Hourly | 0.1° | Global | 1979–NRT |  | Beck et al. (2019b) |
| PERSIANN-CCS | S | Hourly | 0.04° | 60° N/S | 2003–NRT | / | Hong et al. (2004) |
| PERSIANN-CCS-CDR | S,G | 3-hourly | 0.04° | 60° N/S | 1983–2021 | / | Sadeghi et al. (2021) |
| PDIR-Now | S | Hourly | 0.04° | 60° N/S | 2000–NRT | 100 minutes | Nguyen et al. (2020) |
| SM2RAIN-ASCAT | S | Daily | 0.1° | 60° N/S | 2007–2021 | / | Brocca et al. (2019) |
| SM2RAIN-CCI | S | Daily | 0.25° | Global | 1998–2015 | / | Ciabatta et al. (2018) |
| GPM+SM2RAIN | S | Daily | 0.25° | Global | 2007–2018 | / | Massari et al. (2020) |



## 2.2   Streamflow Observations and Catchment Selection

We utilized a comprehensive global database of daily streamflow observations and catchment boundaries compiled from 22 national and international datasets. Appendix A provides a detailed list of the data sources, along with corresponding references and websites. Initially, the database contained 43,627 stations. However, as many stations appeared in multiple data sources, we performed a duplication check and discarded stations where both the station location and the corresponding catchment centroid were within 5 km of those of another station. In case of duplication, regional data sources were prioritized over international

ones (e.g., CAMELS datasets were preferred over GRDC). After this process, the number of unique stations was reduced to 34,768.

To ensure the suitability of the catchments for the present analysis, we applied the following inclusion criteria:

1. Catchment areas were limited to <10,000 km$^2$ to minimize the influence of channel routing, which can become significant at the daily time scale in larger catchments (Gericke and Smithers, 2014). Moreover, since we use catchment-mean $P$

time series to drive the hydrological model, larger catchments are prone to greater spatial averaging, leading to less realistic representation of $P$ patterns.

2. The streamflow record had to be >3 years. This threshold was chosen due to the short records of GDAS, CMORPH-RT, and CMORPH-RAW. We realize that such a short record may introduce some random variability in the KGE scores of these datasets, particularly in arid regions where $P$ events are less frequent. However, this random variability will likely

be averaged out due to the large number of catchments included in our assessment.

3. The number of events (defined as runoff $> 5$ mm d$^{-1}$) had to be $> 10$ non-consecutively, to ensure we have sufficient data for calibration.

4. The mean annual runoff had to be $\geq 5$ and $< 5000$ mm yr$^{-1}$, to filter out catchments with erroneous streamflow and/or catchment boundary data.

5. The reservoir influence (defined as the ratio of total reservoir capacity by mean cumulative annual streamflow) had to be <0.1, as HBV does not explicitly simulate reservoirs. To determine the total reservoir capacity, we used the Global Reservoir and Dam (GRanD) dataset (V1.3; Lehner et al., 2011).

After applying these criteria, 16,295 catchments remained. The 10th, 50th, and 90th percentiles of the catchment areas are 15 km$^2$, 213 km$^2$, and 2688 km$^2$, respectively (Fig. 1).

## 2.3   Hydrological Modeling

The performance of the gridded $P$ datasets was assessed using hydrological modeling for the 16,295 catchments that passed the suitability checks. For each catchment, the HBV conceptual hydrological model (Bergström, 1992; Seibert and Vis, 2012) was calibrated against daily streamflow observations using time series from each $P$ dataset. The HBV model was selected due to its versatility and computational efficiency, and numerous successful applications (see review by Seibert and Bergström, 2022).



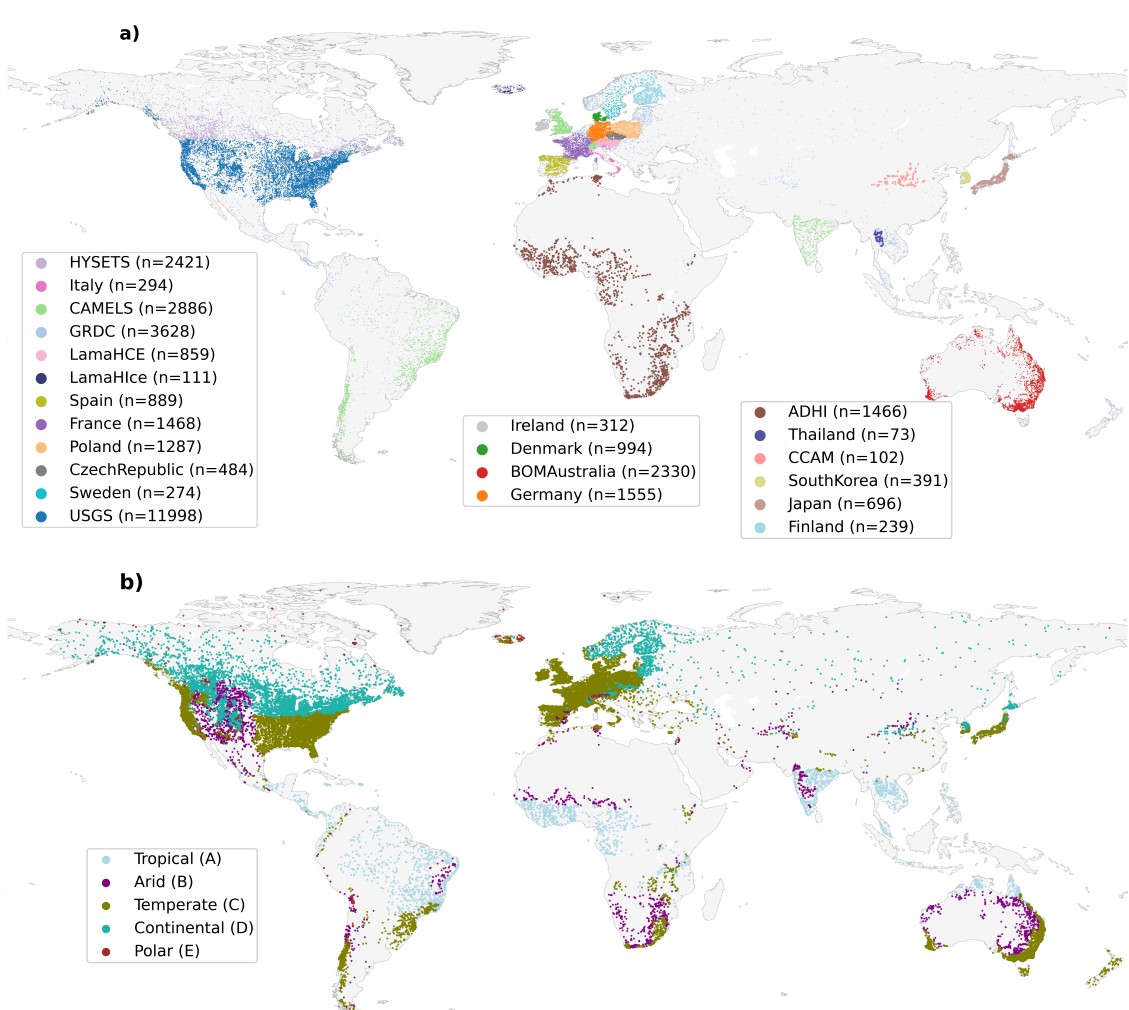

**Figure 1.** Locations of the 34,768 gauges with daily streamflow data that passed the duplication checks, used to evaluate the gridded $P$ datasets. Each data point represents the centroid of a catchment. The colors indicates the dominant major Köppen-Geiger climate class, based on the 1-km resolution map for 1991–2020 from Beck et al. (2023). For more information on the data sources, refer to Appendix A.



The model incorporates two groundwater stores, one unsaturated-zone store, and a triangular weighting function to simulate channel routing delays. Table 2 provides the model parameters and their calibration ranges. An additional parameter, PCORR, was introduced to adjust for systematic biases in $P$ datasets, which are generally easier to mitigate and should, therefore, not disproportionately penalize the datasets.

**Table 2.** HBV model parameter descriptions and calibration ranges.

| Parameter | Description | Minimum | Maximum |
| --- | --- | --- | --- |
| TT (°C) | Threshold temperature when precipitation is simulated as snowfall | -5 | 5 |
| SFCF | Snowfall gauge undercatch correction factor | 1 | 2 |
| CWH | water holding capacity of snowfall | 0 | 0.2 |
| CFMAX (mm °C$^{-1}$ d$^{-1}$) | Melt rate of snowfall | 0.5 | 10 |
| CFR | Refreezing coefficient | 0 | 0.1 |
| FC (mm) | Maximum water storage in unsaturated-zone storage | 50 | 1000 |
| LP | Soil moisture value above which actual evaporation reaches potential evaporation | 0.2 | 1.0 |
| BETA | shape coefficient of recharge function | 1 | 6 |
| UZL (mm) | threshold parameter for extra outflow fromupper zone | 0 | 100 |
| PERC (mm d$^{-1}$) | maximum percolation to lower zone | 0 | 10 |
| K0 (d-1) | Additional recession coefficient of upper groundwater store | 0.005 | 0.9 |
| K1 (d$^{-1}$) | Recession coefficient of upper groundwater store | 0.001 | 0.5 |
| K2 (d$^{-1}$) | Recession coefficient of lower groundwater store | 0.001 | 0.2 |
| MAXBAS | Length of equilateral triangular weighting function | 1 | 10 |
| PCORR | Multiplier to mitigate systematic $P$ underestimation | 1 | 2 |

The model requires daily time series of $P$, potential evaporation, and air temperature as inputs. We used catchment-mean

daily $P$ time series from the gridded datasets listed in Table 1. Potential evaporation was estimated using the Hargreaves (1994) equation, which relies on daily minimum and maximum air temperatures. Catchment-mean daily minimum, maximum, and mean air temperature time series were sourced from the Multi-Source Weather (MSWX) dataset (Beck et al., 2022). A key advantage of MSWX over, for example, ERA5, is its higher resolution (0.1°), which allows for more accurate simulation of snowmelt in mountainous regions.

**2.4  Calibration Procedure**

The 15 model parameters were calibrated for each catchment and $P$ dataset over the period where both observed streamflow and $P$ data were available. Model initialization was done by running the model with 10 years of prior $P$ data, if available. If 10 years of prior $P$ data were not available, the model was run multiple times using the available $P$ data until a total of more than 10 years was accumulated. Calibration was performed using the $(\mu + \lambda)$ evolutionary algorithm (Ashlock, 2010; Fortin et al.,





2012), with a population size ($\mu$) of 20, a recombination pool size ($\lambda$) of 48, and 25 generations, resulting in 1200 model runs per catchment per $P$ dataset, amounting to approximately 19 million model runs in total.

In line with several previous studies (e.g., Beck et al., 2017b; Tarek et al., 2020; Arsenault et al., 2023), we opted not to split the record into separate calibration and validation periods. Instead, the full period of overlapping streamflow and $P$ data was used to maximize the available information for parameter calibration and evaluation and yield more reliable scores. This

is particularly critical for $P$ datasets with short records (e.g., GDAS, CMORPH-RT, and CMORPH-RAW), where splitting the data would lead to scores based on only one or two years of data which could cause instability in the performance score (see Arsenault et al., 2018).

### 2.5 Performance Metric

To assess the performance of streamflow simulations forced by the different gridded $P$ datasets, we calculated the Kling-Gupta

Efficiency (KGE) scores between daily observed and simulated streamflow for each catchment. KGE, introduced by Gupta et al. (2009) and modified by Kling et al. (2012), is an objective performance metric that combines correlation, bias, and variability, and is defined as:

$$\text{KGE} = 1 - \sqrt{(r-1)^2 + (\gamma - 1)^2 + (\beta - 1)^2}, \tag{1}$$

where $r$ represents the Pearson correlation coefficient, $\gamma$ is the ratio of the estimated to observed coefficients of variation, and

$\beta$ is the ratio of estimated to observed means:

$$\gamma = \frac{\sigma_s / \mu_s}{\sigma_o / \mu_o}, \quad \beta = \frac{\mu_s}{\mu_o}, \tag{2}$$

where $\mu$ and $\sigma$ are the mean and standard deviation, respectively, and the subscripts $s$ and $o$ refer to the estimated and observed values. Optimal values for KGE, $r$, $\beta$, and $\gamma$ are all 1. The $r$ term is primarily sensitive to the timing and intensity of $P$ extremes, while $\beta$ captures systematic over- or underestimation of $P$, and $\gamma$ reflects the shape of the $P$ probability distribution.

### 3 Results and Discussion

### 3.1 Overall Model Performance

Fig. 2 presents median calibration scores obtained using the 23 gridded $P$ datasets for the 16,295 catchments. The key findings are summarized below:

– Among the six main categories of $P$ datasets — satellite, gauge, model, satellite+model, satellite+model+gauge, and
satellite+gauge — the satellite category performed the worst overall. This challenges the common assumption among non-experts that satellite datasets, being observation-based and offering high spatial resolution, are inherently better. However, model-based datasets also incorporate observations through the assimilation of extensive surface, radiosonde, and aircraft data. Moreover, a higher spatial resolution does not guarantee better performance, especially when data is



aggregated at the catchment scale. It is worth noting that the catchments in our dataset predominantly represent temperate and cold climates. In tropical regions, satellite datasets often perform better, as discussed in Section 3.2.

– The multi-source dataset MSWEP V2.8 (Beck et al., 2019b) demonstrated the best overall performance. This dataset leverages the complementary strengths of gauge, satellite, and model $P$ estimates to provide improved $P$ estimates across the globe. Specifically, gauge data enhance performance in regions with dense rain gauge networks, satellite estimates enhance performance in convection-dominated regions and periods, and model estimates improve performance in frontal-dominated regions and periods (Beck et al., 2019b).

– Among the purely satellite-based $P$ datasets (CMORPH-RAW and -RT; IMERG-Early and -Late; GSMaP; PDIR-Now; PERSIANN-CCS; and SM2RAIN-ASCAT and -CCI; and GPM+SM2RAIN), the GPM+SM2RAIN dataset (Massari et al., 2020) exhibited the best performance (median KGE of 0.60; Fig. 2). This dataset combines satellite soil moisture retrievals from ASCAT H113 H-SAF, SMOS L3 and SMAP L3 with microwave-based $P$ retrievals from IMERG using the so-called optimal linear combination approach (Bishop and Abramowitz, 2013). IMERG-Late V7 (median KGE of 0.53) introduced several improvements over V6, notably a climatological rain gauge adjustment, leading to a significant performance boost compared to V6 (median KGE of 0.50), particularly in the tropical, cold, and polar catchments (Supplement Fig. S12). In contrast, GSMaP-std V8 dataset (median KGE of 0.44) performed similar to its predecessor, GSMaP-std V7 (median KGE of 0.44).

– Among the purely infrared-based $P$ datasets (PERSIANN-CCS and PDIR-Now), PERSIANN-CCS (Hong et al., 2004) performed better (median KGE of 0.43) than PDIR-Now (Nguyen et al., 2020; median KGE of 0.38). This result is surprising given that PDIR-Now features several improvements over PERSIANN-CCS, such as the dynamic adjustment of the relationship between cloud-top brightness temperatures and rain rates based on rainfall climatologies, as well as the use of a higher temperature threshold to enhance the detection of warm rain events (Nguyen et al., 2020). Further analysis revealed that PDIR-Now performs particularly poorly in the UK, Denmark, and Italy (Supplement Fig. S25), which contributes to its overall poorer performance compared to PERSIANN-CCS.

– Among the purely model-based $P$ datasets (ERA5, GDAS, and JRA-3Q), the recently released reanalysis JRA-3Q, based on the Japan Meteorological Agency (JMA) operational system as of December 2018 (Kosaka et al., 2024), performed best (median KGE of 0.67). GDAS, based on V16.3 from 2022 of the Global Forecasting System (GFS) Numerical Weather Prediction (NWP) model (www.ncei.noaa.gov/products/weather-climate-models/global-forecast), ranked second best (median KGE of 0.63). ERA5, based on Cycle 41r2 of the Integrated Forecasting System (IFS) NWP model from 2016 (Hersbach et al., 2020), yielded the lowest performance (median KGE of 0.59). Although ERA5 is generally considered the most reliable reanalysis for most purposes, these results indicate that JRA-3Q may be a slightly better alternative for hydrological modeling. Note that the GDAS record is much shorter than those of ERA5 and JRA-3Q (Table 1), which substantially limits its applicability.



- Among the rain gauge-based $P$ datasets (CHIRPS 2.0, CPC Unified, GPCP V3.2, IMERG-Final V7, MSWEP V2.8, and PERSIANN-CCS-CDR), MSWEP V2.8 (Beck et al., 2019b) achieved the best overall performance (median KGE of 0.75), underscoring the value of combining $P$ estimates from satellite, model, and gauge data and applying daily gauge corrections. In contrast, CHIRPS V2.0 applies five-day gauge corrections, while the other datasets apply monthly corrections, which provide fewer benefits at the daily time scale. The main challenge in applying daily gauge corrections is the difficulty in accounting for shifts in daily reporting times (i.e., daily accumulations often do not coincide with midnight UTC; Yang et al., 2020). As CPC Unified is solely based on daily gauge observations, its performance is limited by the lack of daily gauge observations in many regions (Kidd et al., 2017). In these regions, the dataset relies entirely on interpolating observations from potentially distant gauges. Another challenge in application of daily gauge corrections is the relatively low coverage of gauge observations in regions outside North America, Europe and Australia.

- The marked differences in median KGE values between MSWEP V2.8 and MSWEP-ng V2.8 (median KGE of 0.75 vs. 0.69), between CHIRPS V2.0 and CHIRP (median KGE of 0.63 vs. 0.57), and between IMERG-Final V7 and -Late V7 (median KGE of 0.67 vs. 0.53) emphasize the importance of applying gauge corrections, in line with previous evaluations (Gochis et al., 2009; Beck et al., 2017b, a; Shen et al., 2018). This highlights the critical role national meteorological agencies play in feeding rain gauge data into global databases such as the Global Historical Climatology Network daily (GHCNd; Menne et al., 2012a) and the importance of improving coverage in data sparse regions due to data sharing limitations.

- A comparison of PCORR parameter values obtained after calibration using different $P$ datasets reveals that the IMERG-Early and -Late V7 datasets necessitate the highest PCORR values, while PDIR-Now is associated with the lowest values (Supplement Figs. S1–S24). The lower PCORR for PDIR-Now reflects its tendency to overestimate $P$, as confirmed by the significant positive bias obtained by the datasets (Fig. 2). This may be because the algorithm was calibrated with a focus on heavy rainfall events for near real-time applications (Nguyen et al., 2020). Conversely, the higher PCORR values required for the IMERG-Early and Late V7 products reflect their tendency to underestimate $P$, which is confirmed by their negative bias values. These differences highlight the variability in performance among satellite $P$ datasets, driven by the unique algorithms utilized in each.

Overall, our findings are consistent with those of Beck et al. (2017b), Gu et al. (2023), and Gebrechorkos et al. (2023), who also evaluated multiple gridded $P$ datasets using hydrological modeling in global catchments. However, while Beck et al. (2017b) assessed nine datasets across 9,053 catchments, Gu et al. (2023) evaluated two datasets across 10,596 catchments, and Gebrechorkos et al. (2023) analyzed six datasets across 1,825 catchments, our study evaluated 23 datasets across 16,295 catchments, making our results more likely to be generalizable. Additionally, Beck et al. (2017b) and Gu et al. (2023) primarily assessed outdated versions of $P$ datasets, whereas our analysis included several new $P$ datasets — such as PDIR-Now, IMERGV7, JRA-3Q, and MSWEP V2.80 — that have not yet been comprehensively evaluated. Furthermore, unlike Gebrechorkos et al. (2023), we recalibrated the hydrological model for each $P$ dataset, which likely reduces potential biases and enhances the reliability of our conclusions.





**Table 3.** Median daily Calibration KGE values for the different $P$ datasets for the five major Köppen-Geiger climate classes. No values are provided for datasets for which the number of calibrated catchments is < 75 % of the total number of catchments. In each column, the dataset with the best performance is shown in bold font. The catchments were classified based on the most dominant class, determined using the 1-km resolution Köppen-Geiger map for 1991–2020 from Beck et al. (2023). See Fig. 1 for a map of the dominant major Köppen-Geiger climate class for the catchments.

| Dataset Type | KG Climate Zone | All | Tropical (A) | Arid (B) | Temperate (C) | Cold (D) | Polar (E) |
|---|---|---|---|---|---|---|---|
| | No. of Catchments | 16295 | 891 | 455 | 11638 | 3152 | 159 |
| | CMORPH-RAW | 0.43 | 0.55 | 0.25 | 0.45 | — | — |
| | CMORPH-RT | 0.52 | 0.55 | 0.28 | 0.51 | — | — |
| | IMERG-Early V7 | 0.53 | 0.60 | 0.35 | 0.51 | 0.58 | 0.69 |
| | IMERG-Late V7 | 0.53 | 0.62 | 0.34 | 0.51 | 0.58 | 0.65 |
| | IMERG-Late V6 | 0.50 | 0.57 | 0.33 | 0.50 | 0.53 | 0.68 |
| S | GSMaP V7 | 0.44 | 0.57 | 0.31 | 0.41 | 0.52 | — |
| | GSMaP V8 | 0.44 | 0.60 | 0.35 | 0.43 | — | — |
| | PERSIANN-CCS | 0.43 | 0.42 | 0.22 | 0.42 | 0.51 | — |
| | PDIR-Now | 0.38 | 0.50 | 0.25 | 0.34 | — | — |
| | SM2RAIN-ASCAT | 0.52 | 0.62 | 0.41 | 0.52 | 0.53 | — |
| | SM2RAIN-CCI | 0.41 | 0.53 | 0.26 | 0.40 | 0.42 | — |
| | GPM+SM2RAIN | 0.60 | 0.64 | 0.44 | 0.60 | 0.60 | — |
| | JRA-3Q | 0.67 | 0.57 | 0.43 | 0.66 | 0.70 | 0.77 |
| M | GDAS | 0.63 | 0.55 | 0.38 | 0.63 | — | — |
| | ERA5 | 0.59 | 0.48 | 0.37 | 0.58 | 0.68 | 0.74 |
| G | CPC Unified | 0.73 | 0.64 | 0.48 | 0.74 | 0.74 | 0.76 |
| | IMERG-Final V7 | 0.67 | 0.66 | 0.48 | 0.67 | 0.71 | 0.74 |
| S+G | PERSIANN-CCS-CDR | 0.43 | 0.45 | 0.26 | 0.42 | 0.50 | — |
| | GPCP V3.2 | 0.66 | 0.65 | 0.47 | 0.65 | 0.70 | 0.77 |
| S+M | CHIRP | 0.57 | 0.55 | 0.33 | 0.55 | 0.66 | — |
| | MSWEP-ng V2.8 | 0.69 | 0.59 | 0.44 | 0.68 | 0.73 | **0.78** |
| S+M+G | CHIRPS V2.0 | 0.63 | 0.63 | 0.42 | 0.61 | 0.70 | — |
| | MSWEP V2.8 | **0.75** | **0.66** | **0.51** | **0.75** | **0.77** | 0.77 |





**Figure 2.** Calibration KGE, correlation ($r$), long-term bias ($\beta$), and variability ratio ($\gamma$) scores achieved by the 23 $P$ datasets. The horizontal orange lines represent the median, the box extends from the 25th to 75th percentiles, while the whiskers represent the 5th and 95th percentiles. The datasets are sorted according to their median KGE values. The colors represent the dataset type: S = Satellite; G = Gauge; M = Model (analysis and reanalysis); S+M = Satellite and Model; S+M+G = Satellite, Model, and Gauge; and S+G = Satellite and Gauge.

## 3.2 Regional Performance Differences

Table 3 presents median calibration KGE scores for the 23 gridded $P$ datasets across the five major Köppen-Geiger climate classes. While satellite $P$ datasets perform the worst overall (see Section 3.1), microwave-based satellite datasets such as IMERG and GSMaP generally outperform model-based datasets (ERA5, GDAS, and JRA-3Q) in tropical regions. This is





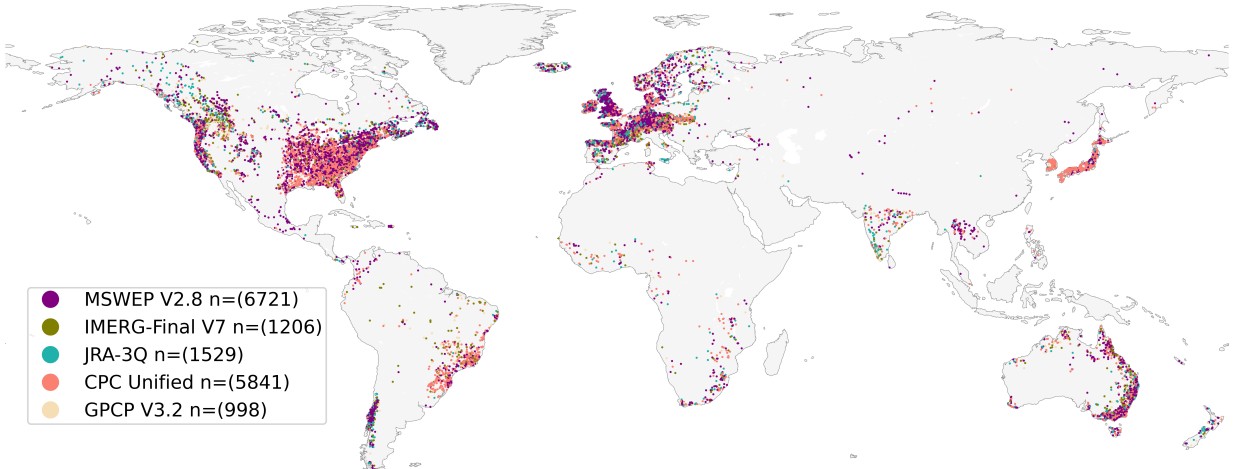

**Figure 3.** The $P$ dataset with the highest calibration KGE for each catchment. Each data point represents the centroid of a catchment (n=16,295). Only the five best-performing $P$ datasets are included. with MSWEP-np V2.8 excluded due to its similarity to V2.8.

likely because tropical $P$ events, typically localized and short-lived, can be directly observed by satellites, while models often
struggle to simulate the complex convective processes driving these events (Yano et al., 2018; Peters et al., 2019; Lin et al., 2022). Conversely, in arid climates, all $P$ datasets tend to perform relatively poorly, with a slight advantage for model-based datasets over satellite-based ones. $P$ in arid regions tends to be brief and intense, making it challenging to detect and model accurately (Beck et al., 2017b; Sun et al., 2018; El Kenawy et al., 2019; Beck et al., 2019a). The occurrence of virga, or $P$ that evaporates before reaching the ground, further complicates accurate estimation in these areas (Wang et al., 2018). In temperate
and, particularly, cold regions, model-based $P$ datasets generally outperform satellite-based datasets, as the large-scale, long-duration frontal $P$ typical of these regions is reliably simulated by models (Ebert et al., 2007; Beck et al., 2017b, 2019a; Sun et al., 2018).

Fig. 4 shows correlations between static catchment attributes (Appendix B) and calibration KGE, correlation ($r$), variability ratio ($\gamma$), and long-term bias ($\beta$) scores for the catchments. We present these correlations for the merged MSWEP V2.8 dataset,
model-based ERA5 dataset, and satellite-based IMERG-Late V7 dataset, shedding light on the ability of different catchment attributes in predicting the performance of each dataset. MSWEP V2.8 and ERA5 exhibit similar results, as ERA5 served as a key input for generating MSWEP V2.8. For MSWEP V2.8, the best predictors of a high KGE are low Mean PET and high Absolute Latitude, likely due to the prevalence of frontal $P$ in these regions, which models simulate well, combined with higher rain gauge densities (Kidd et al., 2017). For ERA5, the best predictors of a high KGE are high Solid P Fraction and low Mean
T, as frontal $P$ is prevalent under these conditions. For IMERG-Late V7, KGE performance is poorly predictable; however, $\beta$ is highly predictable. The best predictors of a low $\beta$ (indicating $P$ underestimation) are low Mean PET and low Mean T, reflecting the satellite's limited ability to detect snowfall (Sadeghi et al., 2019; Song et al., 2021). Contrary to expectations,



Rain Gauge Density did not exhibit strong positive correlations for MSWEP V2.8. Although the Rain Gauge Density map used here may not precisely represent the rain gauges incorporated in MSWEP V2.8, this nonetheless suggests that a high rain
gauge density does not necessarily yield better performance.

Fig. 5 presents median calibration KGE scores obtained by the different $P$ datasets for the different streamflow data sources (see Fig. 1a and Appendix A). Somewhat lower overall performance were obtained for BOMAustralia, CAMELS-INDIA, South Korea, and particularly ADHI. Some discussion on reasons for the lower performance is given below:

– For BOMAustralia (www.bom.gov.au/waterdata/), the lower performance (Fig. 5) is attributed to arid regions exhibiting
consistently low performance (Table 3), with Australian catchments having a particularly high mean aridity index of
   1.5. Additionally, the presence of numerous small dams used for irrigation, domestic water supply, and flood control
   contributes to reduced performance (Ouyang et al., 2021). Our hydrological model, HBV (Bergström, 1992; Seibert
   and Vis, 2012), does not explicitly simulate dams, and although we excluded catchments with significant dam influence
   (see Sect. 2.2), we relied on the GRanD dataset (Lehner et al., 2011), which only includes larger dams. Significant
groundwater withdrawals in Australia, which are not explicitly accounted for by HBV, also impact streamflow.

– For CAMELS-INDIA (Mangukiya et al., 2024), the main data source for India, the lower performance (Fig. 5) is likely
   due to extensive human activity, particularly significant groundwater withdrawals (Rodell et al., 2009; Dangar et al.,
   2021). CAMELS-INDIA catchments have the highest median irrigated area (9.5 %) based on the Global Map of Irrigated
   Areas (GMIA) V5 (Siebert et al., 2005). Additionally, despite excluding catchments with substantial dam influence,
CAMELS-INDIA has the highest median reservoir influence (0.04), so the presence of dams may have further degraded
   performance.

– Similarly, for South Korea (https://water.nier.go.kr), the lower performance (Fig. 5) is likely related to extensive hu-
   man activity, including numerous dams not captured in the GRanD dataset. These dams primarily serve domestic and
   municipal water supplies and irrigation, with catchments having a median irrigated area of 6 % (based on GMIA).

– For ADHI (Tramblay et al., 2021), the main data source for Africa, the arid conditions are likely a major factor for the
   particularly low performance (Fig. 5), given the mean aridity index of 1.5 across these catchments. Another factor to
   consider is the numerous mostly smaller dams across the continent, not included in GRanD and hence not excluded from
   our assessment. Low data quality in flow records may also be a contributing factor, though a global analysis of flow
   data quality does not fully confirm this (Crochemore et al., 2020). Additional challenges for rain gauge-based $P$ datasets
(CHIRPS 2.0, CPC Unified, GPCP V3.2, IMERG-Final V7, MSWEP V2.8, and PERSIANN-CCS-CDR) in Africa in-
   clude low rain gauge density (Kidd et al., 2017), poor data quality, and frequent data gaps. For model-based datasets
   (ERA5, GDAS, and JRA-3), limited availability of surface, radiosonde, and aircraft observations for assimilation in
   Africa contributes to reduced performance (https://charts.ecmwf.int/catalogue/packages/monitoring/). For ERA5 specif-
   ically, the presence of a spurious shift in $P$ in central Africa (also discussed in Zsótér et al., 2020), potentially linked to
the TOVS-to-ATOVS transition, and the occurrence of intense localized rainfall events ("rain bombs") in eastern Africa
   further degrade performance (Hersbach et al., 2020).



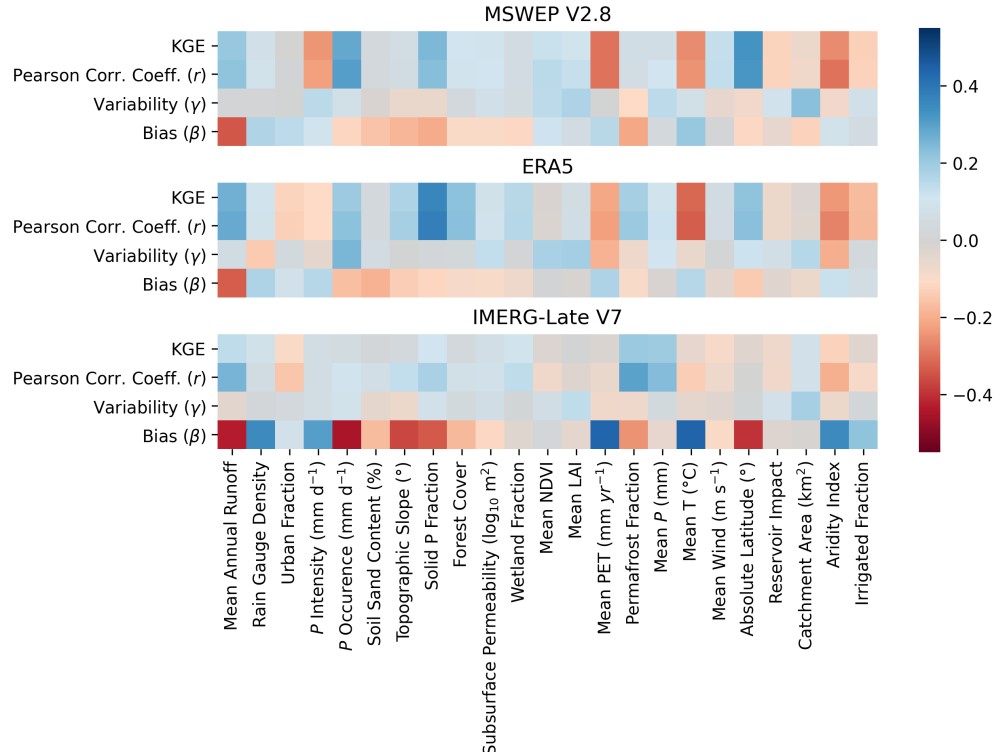

**Figure 4.** Spearman rank correlations between static catchment attributes and calibration KGE, correlation ($r$), long-term bias ($\beta$), and variability ratio ($\gamma$) scores obtained for the catchments using (a) MSWEP V2.8, (b) ERA5, and (c) IMERG-Late V7. See Appendix B for more details on the catchment attributes.

– The low median calibration KGE scores for PDIR-Now in Italy, Denmark, and CAMELS-GB reflect $P$ overestimation, with median bias scores of 1.7, 2.0, and 1.9, respectively, suggesting a tendency of PDIR-Now to overestimate $P$ at higher latitudes. Likewise, the low median calibration KGE of JRA-3Q for Thailand is due to overestimated $P$, with a median bias score of 4.9.

## 3.3 Potential Limitations

We conducted the most extensive evaluation to date of quasi- and fully-global $P$ datasets using hydrological modeling. However, a few potential limitations should be considered when interpreting the results:

1. The calibration process may potentially suppress certain systematic issues inherent in the $P$ datasets, such as consistent under- or overestimation of peaks, long-term biases (SFCF parameter as an example), or the presence of drizzle. As a result, these deficiencies might not be fully captured in our calibration scores. However, this should not necessarily be viewed as a limitation. Systematic issues, once identified, are relatively straightforward to correct through post-





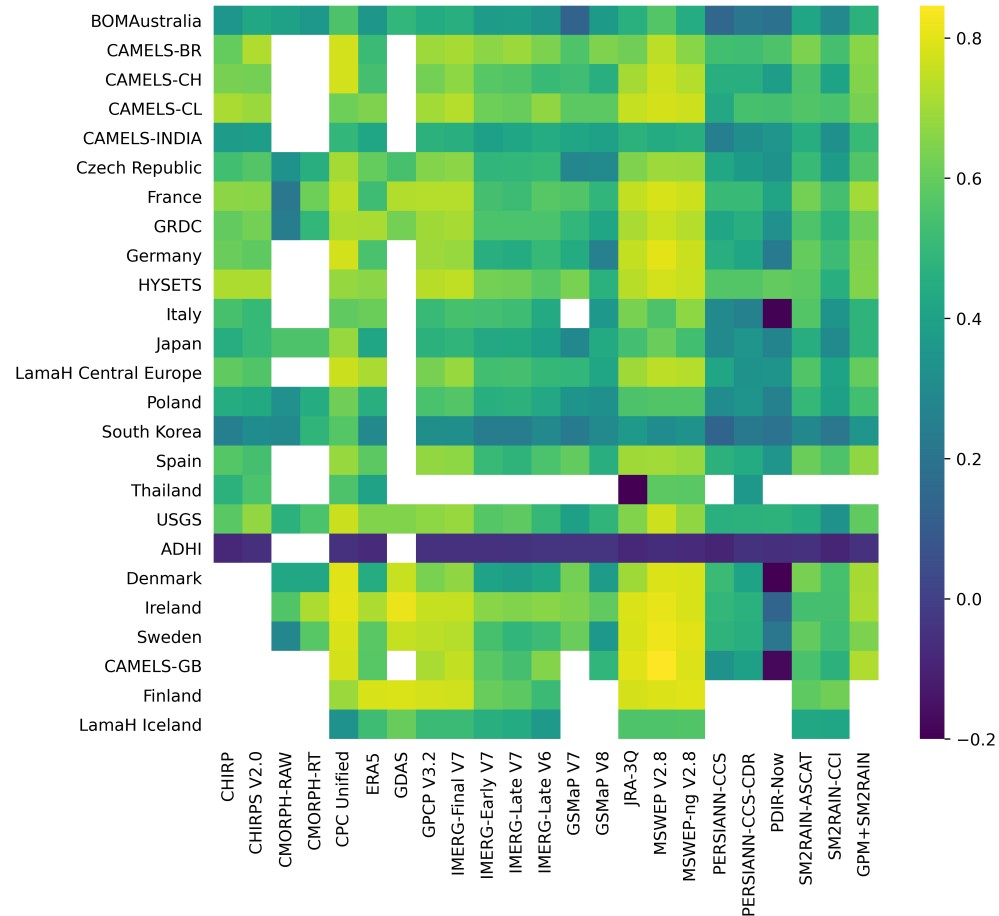

**Figure 5.** Median calibration KGE scores obtained by $P$ datasets for the different streamflow data sources (see Fig. 1a and Appendix A). The white color indicates the absence of data, either due to the unavailability of overlapping streamflow records with the temporal coverage of the $P$ dataset or because no catchments meet the criteria mentioned in section 2.2

processing or bias-adjustment techniques. Consequently, penalizing datasets too heavily for such deficiencies may be unwarranted.

2. While the HBV hydrological model has been widely and successfully applied in numerous studies across various climates and geographic settings (see the review by Seibert and Bergström, 2022), it remains a simple conceptual model with a fixed structure and process representation. As such, other models may provide better simulations (Gu et al., 2023). However, we do not believe that using a different model would alter the relative performance ranking of the $P$ datasets or lead to significantly different conclusions.

3. The HBV model does not explicitly account for human interventions such as dam operations or groundwater withdrawals, which can significantly influence streamflow. However, incorporating human interventions is inherently challenging due



to the lack of consistent and detailed data on water use and management practices. For instance, many large dams, and likely the large majority of smaller ones, are absent from global compilations (Zhang and Gu, 2023), and global sectoral water use data is inherently uncertain, particularly at sub-national scales (e.g., Huang et al., 2018; Puy et al., 2022).

4. We employed a relatively simple temperature-based formulation (Hargreaves, 1994) — which does not explicitly account for the effects of wind speed, radiation, and relative humidity alongside temperature — to estimate potential evaporation. However, we do not believe a more complex formulation such as Penman-Monteith (Penman, 1948; Monteith, 1965) will substantially change the results. This is because streamflow simulations are primarily driven by the $P$ input, and more complex formulations do not necessarily produce more accurate streamflow results (Aouissi et al., 2016; Oudin et al., 2005b, a).

5. We compiled an unparalleled global observed streamflow dataset comprising 34,768 catchments (excluding duplicates) covering all climate zones and latitudes (Fig. 1). Yet, many highly populated and vulnerable regions, particularly in the West Asia, and parts of Central and Eastern Africa, remain underrepresented. This underscores the continued need to improve access to local and regional streamflow data (Krabbenhoft et al., 2022).

6. Since the global distribution of streamflow gauging stations closely aligns with that of meteorological monitoring networks (see Krabbenhoft et al., 2022 and Kidd et al., 2017), our approach may slightly overestimate the relative performance of gauge-based and model-based datasets compared to satellite-only datasets.

7. Some $P$ datasets (GDAS, CMORPH-RT, and CMORPH-Raw) have relatively short record lengths (Table 1), which may have resulted in less reliable KGE scores, particularly in arid regions where $P$ events are less frequent. However, given the large number of catchments included in our assessment, we believe that any potential variability due to these shorter records has been largely eliminated and is unlikely to have affected our main conclusions.

8. Our assessment was carried out on a daily time scale, which obscures critical sub-daily dynamics, particularly in small catchments and arid regions prone to flash floods. Future research may expand our analysis to sub-daily time scales, which would enable a more rigorous evaluation of the timing and intensity of $P$ estimates. Such a sub-daily assessment would likely improve scores for satellite-based $P$ datasets due to their ability to directly observe events, unlike model-based datasets that rely on approximating when such events occur.

## 4 Conclusions

The availability of wide range of gridded $P$ datasets, each with unique technical specifications, strengths, and weaknesses, can make choosing the best dataset for a particular application a complex task. To assist users in making better informed decisions, we conducted the most comprehensive assessment to date of (sub-)daily (quasi-)global gridded $P$ datasets using hydrological modeling. Our analysis involved 23 $P$ datasets evaluated across 16,295 catchments worldwide. For each catchment, we cal-



ibrated a hydrological model using daily streamflow observations, driven by each dataset as input. Our main findings can be summarized as follows:

1. Among the $P$ datasets, MSWEP V2.8 consistently achieved the highest overall performance, owing to its integration of both satellite and model data combined with daily gauge corrections. Satellite datasets performed worst overall. GPM+SM2RAIN performed best among the satellite-based datasets, due to its integration of soil moisture and $P$ retrievals. IMERG-Late V7 showed significant improvements over V6, particularly in tropical and polar regions. Among model-based datasets, JRA-3Q outperformed others, including ERA5, which, despite being one of the most widely used and trusted reanalyses, scored slightly lower in this assessment. MSWEP V2.8 led among the gauge-based datasets, benefiting from its daily gauge corrections, unlike others with five-day or monthly gauge correction. Infrared-based satellite datasets showed lower scores, with PERSIANN-CCS outperforming PDIR-Now.

2. Regional performance of $P$ datasets varied significantly across climates and locations, influenced by local $P$ characteristics, data availability, and human activities. Tropical regions favor microwave-based satellite datasets like IMERG due to their ability to capture localized, convective rainfall, while arid regions exhibited overall poor performance, with model-based datasets slightly outperforming others. In temperate and cold regions, model-based datasets such as JRA-3Q excel in simulating large-scale, frontal $P$. Factors such as aridity, dam presence, and irrigation likely reduced dataset performance in regions like Australia, India, and Africa. The limited availability of in situ meteorological data, combined with potential flow data quality issues, may have further degraded performance in Africa. Specific issues were observed, such as overestimation by PDIR-Now in Europe and JRA-3Q in Thailand.

3. Despite the comprehensiveness of this study, several limitations should be noted. Systematic issues in $P$ datasets may have been partially masked during calibration, though these issues can often be mitigated through post-processing. Additionally, we employed a simple conceptual hydrological model and potential evaporation estimation method, although this is unlikely to have affected the results significantly. The overlap in the global distribution of streamflow and rain gauge networks may have slightly favored gauge- and model-based datasets over satellite-based ones. Lastly, the use of a daily time scale may obscure important sub-daily dynamics, highlighting the need for future sub-daily assessments.

In conclusion, although our findings indicate that datasets like MSWEP V2.8 are well-suited for a broad range of uses, while satellite datasets generally perform worse overall, selecting the most appropriate $P$ dataset ultimately depends on the study region and the specific needs of the application. For example, long-record datasets such as JRA-3Q may be suitable for climate analysis, while IMERG-Early V7 provides a reliable near real-time solution. The continued development of $P$ datasets that balance long-term homogeneity, latency, and spatial-temporal coverage will be essential to meet the varied requirements of users for applications in water resource management, hazard assessment, agriculture, and environmental monitoring.

*Code availability.* The Python code used to generate the results of this study is available from the corresponding author upon request.





## Appendix A: Streamflow Data Sources

370   We compiled an unparalleled database with daily streamflow observations and catchment boundaries for 34,768 catchments
worldwide, drawing from the 22 data sources listed in Table A1. These sources are divided into two categories. The first
category comprises published datasets, including ADHI, HYSETS, CAMELS, LamaHCE, LamaHIce, Germany, and CCAM.
For the remaining sources, except GRDC, daily Observed streamflow data were obtained from the websites of the respective
countries' hydrological or meteorological agencies. Data from GRDC were acquired by submitting an application form on
375   their website and receiving the data via email. For the second set of sources, we used streamflow observations exclusively from
stations with available catchment boundaries, allowing us to calculate time series of meteorological forcings for these catch-
ments, including $P$, temperature, radiation, and humidity. Catchment boundaries for USGS data were sourced from HYSETS,
while those for Italy, Spain, France, Poland, Czech Republic, Sweden, Ireland, Denmark, and Finland came from EStreams
(do Nascimento et al., 2024). For BOM Australia, Thailand, and Japan, boundaries were obtained from GSHA (Yin et al.,
380   2023). The catchment boundaries for South Korea were acquired from Environmental Geographic Information Service, EGIS
(https://egis.me.go.kr/) of South Korea.



**Table A1.** Daily observed streamflow data sources, number of catchments, and references/URLs. The number of stations represents the amount after duplication checks but before suitability checks.

| Data source | Number of stations | Reference/URL |
|---|---|---|
| ADHI | 1466 | Tramblay et al. (2021) |
| BOM Australia | 2330 | www.bom.gov.au/waterdata/ |
| CAMELS | 2887 | Chagas et al. (2020), Höge et al. (2023), Alvarez-Garreton et al. (2018), Coxon et al. (2020), Mangukiya et al. (2024) |
| CCAM | 102 | Hao et al. (2021) |
| Czech Republic | 484 | https://isvs.chmi.cz/ |
| Denmark | 994 | https://odaforalle.au.dk/login.aspx |
| Finland | 239 | wwwi3.ymparisto.fi/i3/paasivu/ENG/Virtaama/Virtaama.htm |
| France | 1469 | www.hydro.eaufrance.fr |
| Germany | 1555 | Loritz et al. (2024) |
| GRDC | 3631 | https://portal.grdc.bafg.de/ |
| HYSETS | 2421 | Arsenault et al. (2020) |
| Ireland | 312 | https://epawebapp.epa.ie/hydronet/#Flow |
| Italy | 294 | www.hiscentral.isprambiente.gov.it |
| Japan | 696 | www.river.go.jp/ |
| LamaHCE | 859 | Klingler et al. (2021) |
| LamaHIce | 111 | Helgason and Nijssen (2024) |
| Poland | 1287 | https://danepubliczne.imgw.pl/ |
| South Korea | 391 | https://water.nier.go.kr/ |
| Spain | 889 | https://ceh.cedex.es/anuarioaforos/demarcaciones.asp |
| Sweden | 274 | www.smhi.se |
| Thailand | 73 | https://hydro.iis.u-tokyo.ac.jp/GAME-T/GAIN-T/routine/rid-river/disc_d.html |
| USGS | 12004 | https://dashboard.waterdata.usgs.gov/app/nwd/en/ |

## Appendix B: Static Catchment Attributes

Table B1 presents the static catchment attributes used for assessing performance predictability. Here, 'static' refers to attributes that do not vary over time. The attributes were calculated for each catchment as described in the table.



**Table B1.** Static catchment parameters, their description and references/URLs.

| Attribute Name | Description |
| --- | --- |
| Mean Annual Runoff | Mean annual runoff (mm yr$^{-1}$) calculated from the observed flow record |
| Rain Gauge Density | Average influence of rain gauges within a catchment. This was calculated by applying a convolution operation to a 0.25-degree global grid. This process utilized an exponential decay kernel of size 10, defined mathematically as $e^{-\alpha\sqrt{x^2+y^2}}$ where $\alpha$ represents the decay rate (set to 1.0). Here, $\sqrt{x^2+y^2}$ calculates the Euclidean distance from the center of the kernel to any given point (x,y) on the grid. The daily rain gauge data employed in this analysis was taken from the Global Historical Climatology Network (GHCN-D; Menne et al., 2012b). This methodology allows for representation of the influence of each rain gauge over its surrounding area, taking into account the natural decrease in influence with increasing distance from each gauge. |
| Urban Fraction | Urban land cover fraction from GlobCover (Bontemps et al., 2011) |
| $P$ Intensity | 99.5th percentile daily $P$ intensity (mm d$^{-1}$) from PPDIST (Beck et al., 2020) |
| $P$ Occurrence | Daily $P$ occurrence (%) using a 0.5 mm d$^{-1}$ threshold from PPDIST (Beck et al., 2020) |
| Soil Sand Content | Soil sand content (%) from SoilGrids250m (Hengl et al., 2017), mean over all layers |
| Topographic Slope | Average slope (%) of the catchment from Global Multi-resolution Terrain Elevation Data 2010 (GMTED2010; Danielson and Gesch, 2011) |
| Solid $P$ Fraction | Fraction of total $P$ falling as snow calculated according to Legates and Bogart Legates and Bogart (2009) using WorldClim V2 (Fick and Hijmans, 2017) for land and ERA5 (Hersbach et al., 2020) for ocean |
| Forest Cover | Forest cover fraction from Food and Agriculture Organization (FAO) Global Forest Resources Assessment (FRA) 2000 (FAO, 2000) |
| Subsurface Permeability | subsurface permeability ($\log_{10}$ m$^2$) from GLobal HYdrogeology MaPS (GLHYMPS) V2.0 (Huscroft et al., 2018) |
| Wetlands Fraction | Wetlands fraction from Global Lakes and Wetlands Database (GLWD) V3 (Lehner and Döll, 2004) |
| Mean NDVI | Normalized Difference Vegetation Index (NDVI) from SPOT-VEGETATION and PROBA-V (Maisongrande et al., 2004) |
| Mean LAI | Mean Leaf Area Index (LAI) from SPOT-VEGETATION and PROBA-V (Fuster et al., 2020) |
| Mean PET | Mean annual potential evapotranspiration (PET) following Consultative Group for International Agricultural Research (CGIAR) V2 (Zomer et al., 2008) |
| Permafrost Fraction | Permafrost fraction following (Brown et al., 1997) |
| Mean $P$ | Mean annual $P$ (mm yr$^{-1}$) from WorldClim V2.1 (Fick and Hijmans, 2017) |
| Mean $T$ | Mean annual air temperature (°C) from WorldClim V2.1 (Fick and Hijmans, 2017) |
| Mean Wind | Mean annual wind speed (m s$^{-1}$) from WorldClim V2.1 (Fick and Hijmans, 2017) |
| Absolute Latitude | Absolute latitude (°) of the centroid of the catchment |
| Catchment Area | Catchment area (km$^2$) |
| Reservoir Impact | Ratio of total reservoir capacity (km$^3$) by annual cumulative streamflow (km$^3$), where the reservoir capacity is taken from Global Reservoir and Dam (GRanD) dataset (V1.3; Lehner et al., 2011) and the annual cumulative flow was calculated from the observed flow record |
| Reservoir Area | Area covered by reservoirs (km$^2$) from Georeferenced global Dams And Reservoirs dataset (GeoDAR) V11 (Wang et al., 2021) |
| Aridity Index | Ratio between mean annual $P$ and potential evapotranspiration, where $P$ was taken from WorldClim V2.1 (Fick and Hijmans, 2017) and potential evapotranspiration from CGIAR V2 (Zomer et al., 2008) |
| Irrigated Fraction | Fraction of irrigated area from Global Map of Irrigated Areas (GMIA) V5 (Siebert et al., 2013) |



385 *Author contributions.* AA: modeling, analysis, visualization, and writing. HB: initial idea, conceptualization, writing, and project administration. All coauthors contributed to writing, revising, and refining the manuscript.

*Competing interests.* The authors declare that they have no conflict of interest.

*Acknowledgements.* We thank the developers of the $P$ datasets listed in Table 1 for their efforts in creating and sharing these valuable resources. Our gratitude also extends to the streamflow data providers listed in Table A1, including the Global Runoff Data Centre (GRDC;
390 Koblenz, Germany), the French National Research Institute for Sustainable Development (RDI), and the Korean National Institute of Environmental Research (NIER). We further thank the developers of the datasets used for the static catchment attributes listed in Table B1. Special thanks are due to Takuji Kubota and Munehisa K. Yamamoto for their valuable insights regarding the performance of GSMaP. Part of the analysis was conducted using Shaheen III, managed by the Supercomputing Core Laboratory at King Abdullah University of Science & Technology (KAUST) in Thuwal, Saudi Arabia.



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
