# Peer review of "Comprehensive Global Assessment of 23 Gridded Precipitation Datasets Across 16,295 Catchments Using Hydrological Modeling"

_EGUsphere, 2024_

## Author Comment (AC1)

This article presents a comprehensive evaluation of the daily performance of 23 global precipitation (P) datasets through a hydrological modeling experiment using the HBV model across 16,295 catchments worldwide. The manuscript is very well-written and clear, and the study addresses a topic of interest to the scientific community. It contributes valuable insights into the suitability of different P datasets for hydrological applications at the global scale and fits well within the scope of the journal. Overall, this is a strong and useful contribution, and I congratulate the authors for the scale and depth of the analysis. I believe that considering the following points will further enhance the manuscript.

**Response:** Thank you very much for your time to review the manuscript. We have revised the manuscript in accordance with your suggestions. We would also like to mention that we have used a revised and updated streamflow database, which includes a higher number of data sources (increased from 22 to 29), improved temporal coverage, and bug fixes—particularly for streamflow records from Africa (ADHI). The extended temporal coverage has led to an increase in the number of catchments with sufficient data for model calibration. As a result, the total number of catchments for which HBV parameters were optimized against observed streamflow increased from 16,295 to 18,428. We have also removed CMORPH-RAW and included CMORPH-CDR, following the recommendation of Reviewer 1. While the overall findings remain consistent with those in the previous version, we have updated the manuscript wherever new insights emerged. Please note that the line numbers and figure numbers in our responses correspond to the updated manuscript

I understand the rationale behind calibrating both the snowfall gauge undercatch correction factor (SFCF) and the multiplicative bias correction factor (PCORR), as these systematic errors can be addressed more easily. However, I suggest lowering the minimum bound of these parameters (e.g., to 0.6) to avoid favouring datasets that tend to underestimate P. In addition, it would be helpful to compare the results of the current calibration approach with a scenario where SFCF and PCORR are both fixed at 1.0. This comparison could shed light on the overall performance of P datasets. More specifically about (i) which datasets tend to systematically over- or underestimate and where, and (ii) the relative importance of these biases when these products are used for hydrological modelling purposes.

**Response:** We evaluated the performance of the precipitation datasets by calibrating KGE values under four different scenarios. In the first two scenarios, PCORR was allowed to vary between 0.0–2.0 and 0.5–2.0, respectively. In the third scenario, both PCORR and SFCF were fixed at 1.0, as suggested by the reviewer. The fourth scenario reflects our original setup, where both PCORR and SFCF were allowed to vary between 1.0 and 2.0. The resulting rankings of the precipitation datasets based on KGE values are illustrated in the figure below.

[Figure]

**Fig S26.** Kling-Gupta Efficiency values for P datasets in different scenarios of PCORR and SFCF calibration.

These results indicate that the overall ranking of precipitation datasets does not vary significantly across the four scenarios. However, in scenario three (PCORR: 1.0, SFCF: 1.0), the rankings of GPCP V3.2 and ERA5 improved. Notably, this improvement in ranking is not due to higher KGE values, as the overall KGE in this scenario is lower compared to the other three. The figure below illustrates the decrease in KGE when PCORR and SFCF were fixed during calibration (scenario three), compared to the default scenario where these parameters were allowed to vary between 1.0 and 2.0.

[Figure]

**Fig.** Decrease in KGE values when PCORR and SFCF were kept constant during calibration (1.0) as compared to when they were allowed to vary between 1.0 to 1.0.

The distribution of PCORR values across the four calibration scenarios (as shown in the following figure) reveals that only a few precipitation datasets exhibit significant adjustment below 1.0 when PCORR is allowed to vary from 0.0 to 2.0. Specifically, PDIR-Now, GSMaP V7, PERSIANN-CCS-CDR, and IMERG-Late V6 show a notable portion of calibrated PCORR values falling below 1.0. This indicates that, for these datasets, the raw precipitation input tends to overestimate actual precipitation in a substantial number of catchments.

[Figure]

**Fig S27.** Comparison of distribution of calibrated PCORR values from four calibration scenarios.

We have added the following sentences in our manuscript to describe this.

Lines 137 - 144: *"To assess the influence of systematic P bias correction using the PCORR and SFCF adjustment factors on model performance, we explored four calibration scenarios with varying bounds for the PCORR and SFCF parameters. In the first scenario, PCORR was allowed to vary between 0.0 and 2.0, providing full flexibility to adjust for both under- and overestimation of P, while SFCF was allowed to vary between 1.0 to 2.0. The second scenario limited PCORR to the range 0.5–2.0, while keeping the range of SFCF between 1.0 and 2.0. The third scenario fixed both PCORR and SFCF parameters at 1.0, effectively disabling P bias correction. The fourth scenario constrained both PCORR and SFCF to the range 1.0–2.0, allowing only upward correction. These scenarios enabled us to evaluate the sensitivity of model performance to P bias correction and assess the robustness of P dataset rankings under varying calibration constraints."*

Lines 239 - 245: *"The overall ranking of P datasets remained largely consistent across the four PCORR calibration scenarios (Supplement Fig. S26). However, in the scenario where PCORR and SFCF were fixed at 1.0, GPCP V3.2 and ERA5 showed improved relative rankings—not due to higher performance, but because other datasets experienced greater performance drops under this constraint. Most datasets showed little sensitivity to the PCORR bound below 1.0, but a few—namely PDIR-Now, GSMaP V7, PERSIANN-CCS-CDR, and IMERG-Late V6—exhibited notable use of PCORR values below 1.0 (Supplement Fig. S27). This suggests that these datasets tend to overestimate P in certain catchments, and that downward rescaling improves their hydrological performance".*

The HBV model is applied in a lumped configuration, considering catchment-averaged forcing data. While this is understandable for a global-scale study, it would be very interesting to explore whether a semi-distributed configuration that accounts more explicitly for P gradients related to elevation, could provide additional insights, particularly in mountainous or topographically complex regions.

It would also be helpful to include a short statement in the Limitations Section acknowledging the assumption of constant land cover over the analysis period. Land cover changes can influence hydrological responses and might introduce some uncertainties in the model performance, especially over multi-decadal periods.

**Response:** Thank you for highlighting this important point. You are correct that in this study we did not subdivide each catchment into sub-catchments or sub-basins; instead, we used catchment-averaged forcing data to drive the HBV model. While using a semi-distributed model configuration with elevation bands would better account for spatial variability in precipitation and temperature—particularly in mountainous or snow-influenced catchments—implementing such an approach would significantly increase computational demand, especially at the global scale of this study. We therefore considered this beyond the scope of the current work, which focuses on providing a first large-scale comparative assessment of precipitation datasets. Nonetheless, we fully agree that exploring the use of elevation bands would be a valuable direction for future research to further improve hydrological simulations in complex terrain.

We also did not consider land-use and land cover changes with time which can occur when the simulation period is long. However, we do not think this will significantly alter the findings from this study i.e. regarding the suitability of P datasets for different regions. We have added both limitations to the 'Potential Limitations and Future Work' section of the manuscript.

Lines 343 – 347: "*Additionally, it does not account for spatio-temporal variations in land cover or use and relies on catchment-averaged meteorological forcings, omitting sub-catchment variability in climate and terrain. More complex (semi-)distributed models with hydrologic response units or elevation bands may yield improved simulations (Gu et al., 2023). However, we do not expect this to materially affect the relative performance ranking of the P datasets or the main conclusions.*".

Since PCORR and SFCF are calibrated, I also recommend including a few sentences explaining which types of biases are captured by the beta component of the KGE. This would clarify the references throughout the manuscript to over- and underestimation of datasets, which in part, could be related to biases of different magnitudes for specific P intensities, and the skill of the products to accurately detect P events.

**Response:** Thank you for this helpful comment. The β component of KGE reflects the ratio of the mean simulated to mean observed streamflow and thus captures systematic biases in the volume of simulated streamflow relative to observations. While PCORR and SFCF parameters of the HBV model are calibrated to correct for overall biases in precipitation and snowmelt inputs, residual biases may remain due to limitations in the precipitation product's ability to correctly represent the spatial and temporal distribution of P intensities and magnitudes (Sun et al., 2017). These residual biases can still influence streamflow and are partially reflected in the β term of KGE. We have clarified this in the revised manuscript.

Lines 161 – 164: "*While the PCORR and SFCF parameters, which account for systematic biases, were calibrated, the β component of KGE reflects residual biases that may persist due to limitations in the P dataset's ability to accurately represent the spatial and temporal distribution of precipitation intensities and magnitudes (Sun et al., 2017)*".

In Section 2.2, the authors mention that streamflow records of selected catchments must span more than three years. Could the authors clarify if these years must be consecutive? Similarly, the rationale for requiring more than 10 non-consecutive P events is not fully explained. How was this threshold determined?

**Response:** One of the requirements for a catchment to be considered or not was to have at least 3 years total observation streamflow record. This record needs not to be consecutive.

Line 95: "*The total streamflow record had to be >3~years, not necessarily consecutive.*"

Additional minor suggestions:

**Response:** Thank you for your suggestions. We have implemented them in our updated manuscript.

Table 1: In the "Temporal resolution" column please use either "30 min." or "30 min" consistently.

**Response:** Thank you for highlighting this issue. We have uniformly used 'min.' in Table 1 for temporal resolution.

L156: It would be helpful to report the median KGE for MSWEP V2.8 here for easier comparison with other products.

**Response:** The overall median KGE value for MSWEP V2.8 is 0.75. We have mentioned this value in that sentence.

Lines 176 – 177: "The multi-source MSWEP V2.8 dataset (Beck et al., 2019b) demonstrated the best overall performance (median KGE of 0.78)."

L189: Likewise, indicate the median KGE for CHIRPS V2.0.

**Response:** The overall median KGE value for CHIRPS V2.0 is 0.66. We have modified the sentence to include this value.

Lines 208 – 209: ". In contrast, CHIRPS V2.0 (median KGE of 0.66) applies five-day gauge corrections, while the other datasets apply monthly corrections, which provide fewer benefits at the daily time scale."

Table 3: IMERG-Final V7 also performs best over tropical regions and should be marked in bold, as is done for MSWEP V2.8.

**Response:** You are correct that the median KGE values for both IMERG-Final V7 and MSWEP V2.8 in tropical regions were 0.66. However, we had highlighted only MSWEP V2.8 due to its coverage of a larger number of catchments compared to IMERG-Final V7. In the revised Table 3, we have now included the number of catchments for each dataset within each climate zone. The first row shows the total number of catchments, but since the availability of precipitation data varies across datasets, the actual number of catchments used for each dataset also differs.

Throughout the manuscript, "evaporation" and "evapotranspiration" are used interchangeably. Please consider clarifying and using these terms consistently.

**Response:** Thank you for highlighting this issue. We used the term "evaporation" instead of "evapotranspiration" based on the recommendation by Miralles et al. (2020). Additionally, Seibert and Vis (2012) also use the term "evaporation" when describing the structure of the HBV model. To maintain consistency, we have replaced the two instances of "evapotranspiration" with "evaporation" in the manuscript.

**References:**

Miralles, D. G., Brutsaert, W., Dolman, A. J., & Gash, J. H. (2020). On the use of the term "evapotranspiration". *Water Resources Research*, 56(11), e2020WR028055. https://doi.org/10.1029/2020WR028055

Seibert, J. and Vis, M. J. P.: Teaching hydrological modeling with a user-friendly catchment-runoff-model software package, Hydrology and Earth System Sciences, 16, 3315–3325, 2012.

Sun, Q., Miao, C., Duan, Q., Ashouri, H., Sorooshian, S., & Hsu, K. L. (2018). A review of global precipitation data sets: Data sources, estimation, and intercomparisons. *Reviews of Geophysics*, *56*(1), 79-107.

---

## Author Comment (AC2)

Review of EGUsphere-2024-4194

Title

Comprehensive Global Assessment of 23 Gridded Precipitation Datasets Across 16925 Catchments Using Hydrological Modeling

By: Ather Abbas, Yuan Yang, Ming Pan, Yves Tramblay, Chaopeng Shen, Haoyu Ji, Solomon H. Gebrechorkos, Florian Pappenberger, JonCheol Pyo, Dapeng Feng, George Huffman, Phu Hguyen, Christian Massari, Luca Brocca, Tan Jackson, & Hylke E. Beck

This manuscript provides an extensive evaluation of the hydrological performance of 23 gridded precipitation (P) datasets by calibrating a hydrological model over 16,295 catchments across the globe. The 23 P datasets are chosen based on their availability at (sub)-daily scale and their (quasi)-global coverage. Among them, 1 dataset is gauge-based only, 3 are reanalysis-based only, 12 are satellite-based only, 3 combine gauge and satellite data, 2 combine reanalysis and satellite data, and 2 combine gauge, reanalysis and satellite data. A conceptual hydrological model (HBV) is used to simulate daily streamflow and is calibrated with each P dataset using the evolutionary algorithm. The Kling-Gupta Efficiency (KGE) is used to assess the hydrological performance of each P dataset across 16,295 catchments.

**Response:** Thank you very much for your time to review our manuscript and your thoughtful comments. We have revised the manuscript in accordance with your suggestions. We would also like to mention that we have used a revised and updated streamflow database, which includes a higher number of data sources (increased from 22 to 29), improved temporal coverage, and bug fixes—particularly for streamflow records from Africa (ADHI). The extended temporal coverage has led to an increase in the number of catchments with sufficient data for model calibration. As a result, the total number of catchments for which HBV parameters were optimized against observed streamflow increased from 16,295 to 18,428. We have also removed CMORPH-RAW and included CMORPH-CDR, following the recommendation of Reviewer 1. While the overall findings remain consistent with those in the previous version, we have updated the manuscript wherever new insights emerged. Please note that the line numbers and figure numbers in our responses correspond to the updated manuscript.

General Comments:

This study tackles a crucial issue in hydrological modeling, which is becoming increasingly important for many potential users who often lack guidance and confidence when navigating the wide range of products available for addressing specific

hydrological problems across different geographical regions. The manuscript is well-written and well-structured; however, several issues need to be addressed to strengthen the robustness of the findings and conclusions.

**Response:** Please find the point-by-point response below.

The inclusion criteria stated in Section 2.2 (L80-104) are questionable because of their subjective nature and arbitrary setting. To ensure sufficient data for calibration, the number of events (defined as runoff > 5 mm d$^{-1}$) must exceed 10 non-consecutively (L96-97). However, some fixed values were set by the authors without providing a clear explanation of their rationale behind their selection. Would it be more appropriate to use a percentile of runoff instead of a fixed value for adaptation across catchments with different hydrological regimes? Similarly, to filter out catchments with erroneous streamflow and catchment boundary data, the authors set the mean annual runoff to be ≥ 5 and < 5000 mm yr$^{-1}$ (L98-99). However, the range of mean annual runoff values can vary significantly across different climatic zones, with arid regions ranging from 0 to 100 mm yr$^{-1}$, while tropical regions can vary from 800 to over 2000 mm yr$^{-1}$. It would be appreciated if the authors could provide further explanation and justification for their inclusion criteria.

**Response:** Thank you for raising this important point. We agree that our inclusion criteria are subjective and may raise questions. However, for each criteria, we clearly explained the purpose it serves in the text. Additionally, choosing different criteria is unlikely to result in a different performance ranking of P datasets or different conclusions. Regarding the criteria of >10 events of >5 mm/d, this is to ensure we have sufficient dynamic events for calibration, and we think this criteria is not controversial, and is suitable for all climate zones, including arid ones. We are not convinced about using percentiles, as this might unrealistically identify very small streamflow occurrences as events, in case a runoff record is largely devoid of any runoff. Regarding the mean annual runoff threshold, in our experience <5 mm/yr or >5000 mm/yr of runoff is extremely rare, and therefore this criteria is useful to identify catchments with erroneous data, for example due to a mistaken unit conversion. We do not think this criteria is controversial or requires a longer justification than we already have.

The hydrological performance of P datasets with higher spatial resolution might be compromised when using catchment-mean P to drive the hydrological model, as the more detailed spatial information from these P datasets is lost. It is somewhat surprising to see that CPC Unified ranks second, given its coarse spatial resolution (0.5°). In addition, it is quite interesting that JRA-3Q performs better than its higher spatial resolution counterparts (ERA5 and GDAS). It is suspected that using catchment-mean P might mask the advantages of higher spatial resolution, leading to the conclusion that "higher spatial resolution does not guarantee better performance, especially when data

is aggregated at the catchment scale" (L153-154). This may hold true for catchments dominated by a single climate or with relatively uniform topography, where spatial variability in precipitation has less influence. However, in mountainous, snow-dominated, or mixed-climatic catchments, the hydrological response cannot be adequately captured without detailed spatial P information. As a result, the true value of higher spatial resolution datasets may be underestimated, potentially biasing the selection of P datasets for hydrological modelling.

**Response:** Thank you for your comment. The usage of catchment averaged forcing data is one of the limitations of current work which can diminish the advantage of high-resolution P datasets over lower resolution datasets especially for large catchments. One way of overcoming this limitation is dividing the catchment into sub-basins, for example based upon elevation bands. While using a semi-distributed model configuration with elevation bands would better account for spatial variability in precipitation and temperature—particularly in mountainous or snow-influenced catchments—implementing such an approach would significantly increase computational demand, especially at the global scale of this study, and we are quite confident it won't change the performance ranking of the P datasets and our main conclusions. We therefore considered this beyond the scope of the current work. Nevertheless, we have added this point in "Potential Limitations and Future Work" section of our manuscript.

Lines 343 – 347: "*Additionally, it does not account for spatio-temporal variations in land cover or use and relies on catchment-averaged meteorological forcings, omitting sub-catchment variability in climate and terrain. More complex (semi-)distributed models with hydrologic response units or elevation bands may yield improved simulations (Gu et al., 2023). However, we do not expect this to materially affect the relative performance ranking of the P datasets or the main conclusions.*"

The use of PCORR parameter to mitigate systematic biases in P datasets during calibration may present challenges because it adjusts only for P underestimation by setting the range between 1 to 2. This focuses on correcting underestimation without addressing P overestimation could disproportionately affect datasets prone to overestimation, potentially skewing performance evaluations. For instance, datasets like PDIR-Now and JRA-3Q, which experience overestimation, have low median KGE scores in some streamflow data sources (L277-280). It would be appreciated if the authors could provide a more comprehensive explanation and justification for their focus on mitigating only P underestimation.

**Response:** We evaluated the performance of the precipitation datasets by calibrating KGE values under four different scenarios. In the first two scenarios, PCORR was allowed to vary between 0.0–2.0 and 0.5–2.0, respectively. In the third scenario, both PCORR

and SFCF were fixed at 1.0, as suggested by reviewer 3. The fourth scenario reflects our original setup, where both PCORR and SFCF were allowed to vary between 1.0 and 2.0. The resulting rankings of the precipitation datasets based on KGE values are illustrated in the figure below.

[Figure]

**Fig. S26.** Kling-Gupta Efficiency values for P datasets in different scenarios of PCORR and SFCF calibration.

These results indicate that the overall ranking of precipitation datasets does not vary significantly across the four scenarios. However, in scenario three (PCORR: 1.0, SFCF: 1.0), the rankings of GPCP V3.2 and ERA5 improved. Notably, this improvement in ranking is not due to higher KGE values, as the overall KGE in this scenario is lower compared to the other three. The figure below illustrates the decrease in KGE when PCORR and SFCF were fixed during calibration (scenario three), compared to the default scenario where these parameters were allowed to vary between 1.0 and 2.0.

[Figure]

**Fig.** Decrease in KGE values when PCORR and SFCF were kept constant during calibration (1.0) as compared to when they were allowed to vary between 1.0 to 1.0.

The distribution of PCORR values across the four calibration scenarios (as shown in the following figure) reveals that only a few precipitation datasets exhibit significant adjustment below 1.0 when PCORR is allowed to vary from 0.0 to 2.0. Specifically, PDIR-Now, GSMaP V7, PERSIANN-CCS-CDR, and IMERG-Late V6 show a notable portion of calibrated PCORR values falling below 1.0. This indicates that, for these datasets, the raw precipitation input tends to overestimate actual precipitation in a substantial number of catchments.

[Figure]

**Fig S27.** Comparison of distribution of calibrated PCORR values from four calibration scenarios.

We have added the following sentences in our manuscript to describe this.

Lines 137 - 144 : *"To assess the influence of systematic P bias correction using the PCORR and SFCF adjustment factors on model performance, we explored four calibration scenarios with varying bounds for the PCORR and SFCF parameters. In the first scenario, PCORR was allowed to vary between 0.0 and 2.0, providing full flexibility to adjust for both under- and overestimation of P, while SFCF was allowed to vary between 1.0 to 2.0. The second scenario limited PCORR to the range 0.5–2.0, while keeping the range of SFCF between 1.0 and 2.0. The third scenario fixed both PCORR and SFCF parameters at 1.0, effectively disabling P bias correction. The fourth scenario constrained both PCORR and SFCF to the range 1.0–2.0, allowing only upward correction. These scenarios enabled us to evaluate the sensitivity of model performance to P bias correction and assess the robustness of P dataset rankings under varying calibration constraints."*

Lines 239 - 245 : *"The overall ranking of P datasets remained largely consistent across the four PCORR calibration scenarios (Supplement Fig. S26). However, in the scenario*

*where PCORR and SFCF were fixed at 1.0, GPCP V3.2 and ERA5 showed improved relative rankings—not due to higher performance, but because other datasets experienced greater performance drops under this constraint. Most datasets showed little sensitivity to the PCORR bound below 1.0, but a few—namely PDIR-Now, GSMaP V7, PERSIANN-CCS-CDR, and IMERG-Late V6—exhibited notable use of PCORR values below 1.0 (Supplement Fig. S27). This suggests that these datasets tend to overestimate P in certain catchments, and that downward rescaling improves their hydrological performance".*

Specific Comments:

L57-60: It would be appreciated if the authors could provide some basic information about the new datasets in the Gridded P Datasets section 2.1.

**Response:** Table 1 summarizes basic information of each of the 23 P datasets including data source (satellite, (re)analysis, gauge), spatial and temporal resolution, spatial and temporal coverage, time latency, full name as well as the reference.

L121-124: It is very unclear that how the model was initialized when 10 years of prior P data were not available. Did the authors just concatenate the same available P data n times to achieve the desired length? Or did the authors use any rainfall generators to produce stochastic P data? Please clarify and justify the use of "multiple times using the available P data until a total of more than 10 years was accumulated"?

**Response:** Yes, we concatenated the forcing data to achieve the desired length. This is because several years of initialization is necessary to bring the stores to optimum level. We would also like to highlight that even if we initialized the model by running it for 10 years, we still did not use the simulated results from the first 365 days to calculate performance during calibration. We have further clarified this in our manuscript.

L124-126: It would be appreciated if the authors could provide more information and description about the evolutionary algorithm.

**Response:** We used an Evolutionary Algorithm (EA) to calibrate the HBV model parameters. EA is a population-based optimization method where each individual represents a potential parameter set, evaluated using the Kling-Gupta Efficiency (KGE) and constrained by physically meaningful parameter ranges. The algorithm evolves the population across generations through biologically inspired operations such as crossover and mutation. In our implementation, 90% of the offspring were generated using blend crossover and 10% through Gaussian-based mutation. Offspring were then evaluated by running the HBV model with the new parameters. This process was repeated until a stopping criterion was met—either reaching the maximum number of generations or detecting convergence, defined as no improvement in the best KGE score greater than

0.0001 over 10 consecutive generations. We have added more description about it in the main manuscript.

Lines 136 - 139: *"We used a (µ + λ) evolutionary algorithm, which is a population-based optimization method that iteratively evolves solutions through selection, crossover, and mutation to maximize the Kling-Gupta Efficiency (KGE) objective. The algorithm was implemented using version 1.4 of Distributed Evolutionary Algorithms in Python (DEAP) library (Ashlock, 2010; Fortin et al., 2012), with a population size (µ) of 20, a recombination pool size (λ) of 48, crossover of 90% and Gaussian based mutation of 10%. To ensure convergence, the optimization process was terminated if the best KGE value did not improve by more than 0.01 for 5 consecutive generations after a minimum of 25 generations."*

L128-132: For a particular catchment, the full period of overlapping streamflow and P data could be different because of the differences in the temporal availability of the P datasets. In this regard, will such differences also cause instability in the performance score?

**Response:** Thank you for this insightful comment. The temporal range of data for each catchment is different from each other. This is due to the difference in availability of streamflow records for different catchments and availability of P datasets. Unfortunately using a consistent time period for each P dataset would lead to exclusion of certain datasets from the analysis. However, since the number of catchments are very large, we believe the median performance metrics presented in our study for each region will not vary if a constant temporal range for each P dataset were used.

L170-176: Will the poor performance of PDIR-Now due to the inability of PCORR in adjusting overestimation of the P dataset?

**Response:** Thank you for your observation. This is partially correct in the case of PDIR-Now. We analyzed the effect of four calibration configurations for the PCORR and SFCF parameters in the HBV model, applied across all 23 P products and all stations. The table below summarizes the performance (in terms of KGE) and the median calibrated PCORR and SFCF values for PDIR-Now under each scenario:

| Scenarios | KGE | PCORR | SFCF |
|---|---|---|---|
| PCORR: 0–2, SFCF: 1–2 | 0.49 | 1.08 | 1.39 |
| PCORR: 0.5–2, SFCF: 1–2 | 0.48 | 1.10 | 1.4 |
| PCORR: 1.0 (fixed), SFCF: 1.0 (fixed) | 0.42 | 1.0 | 1.0 |
| PCORR: 1.0–2.0, SFCF: 1–2 | 0.45 | 1.17 | 1.36 |

The performance improved in the first two scenarios (median KGE = 0.49 and 0.48, respectively) when PCORR was allowed to vary below 1.0. This added flexibility enabled the model to slightly reduce precipitation where needed, suggesting that PDIR-Now may overestimate precipitation in some catchments. Therefore, your observation is valid: the lower performance of PDIR-Now can be partially attributed to the restriction of PCORR to values ≥ 1.0, which limits its ability to adjust for overestimated precipitation. Allowing PCORR to vary below 1.0 partially improves model performance by enabling both upward and downward corrections. However, this pattern does not hold for all precipitation datasets. The figure below shows the improvement in KGE when the PCORR calibration range was expanded from 1.0–2.0 (scenario 4) to 0.0–2.0 (scenario 1). The most notable improvement is observed for GSMaP V7, with an increase in KGE of 0.06, followed by PDIR-Now, which shows an improvement of 0.04. For most of the remaining precipitation datasets, the change in KGE was marginal, with improvements generally around 0.01, indicating limited benefit from allowing PCORR values below 1.0.

[Figure]

**Fig S28.** Improvement in KGE values for different P datasets by increasing the range of PCORR from 1.0-2.0 to 0.0 to 2.0 during calibration.

We have added the following lines in section 3.1 of manuscript

Lines 247 – 253: "*The lower performance of PDIR-Now can partially be attributed to the inability of PCORR in adjusting overestimation of the P. This is evident from lower calibrated values of PCORR when its value was allowed to vary below 1.0. The median calibrated PCORR value decreased from 1.2 to 1.1 which also resulted in a slight improvement in median KGE value from 0.43 to 0.47. Further analysis revealed that the largest decrease in median calibrated PCORR (1.0 to 0.7) and consequently improvement in KGE (0.15 to 0.37) was observed in CAMELS-GB (Supplement Fig. S29). However, a comparison of all P datasets indicates that improvement in KGE for most other P datasets was insignificant, affirming that the range of PCORR (1.0–2.0) is reasonable for most P datasets (Supplement Fig. S28).*"

L310-312: Could the authors elaborate further how the alignment of streamflow stations with meteorological network might favour gauge-based and reanalysis-based P datasets over satellite-only P datasets?

**Response:** This is primarily because gauge-based P datasets incorporate gauge data and reanalysis-based P datasets assimilate temperature, humidity, and other observations in these regions. Since streamflow gauges are typically located in regions with denser meteorological networks, these datasets tend to perform better in such areas. As a result, when performance is evaluated using streamflow data from these regions, it may lead to a slight overestimation of the relative performance of gauge-based and reanalysis-based datasets compared to satellite-only datasets, which do not benefit from local gauge input.

Remarks:

L54: typo "result n biased conclusions"

**Response:** Thank you for highlighting the typo. We have corrected this in the updated manuscript.

Lines 55 – 57: "*Additionally, several studies failed to re-calibrate the hydrological model for each P dataset, including the recent global assessment by Gebrechorkos et al. (2023), which could result in biased conclusions.*"

L72: should it be "IMERG-Early V7" instead of "IMERG-Early V6"?

**Response:** Thank you. Yes, we have used IMERG-Early V7 and not IMERG-Early V6 in this study. We have corrected the sentence.

L275: please explain "TOVS-to-ATOVS transition. Thank you.

**Response**: The TOVS-to-ATOVS transition in ERA5 refers to a change in satellite observation systems. TOVS (TIROS Operational Vertical Sounder) was an older generation of sounders used from the 1970s to the 1990s. In 1998–1999, it was replaced by the more advanced ATOVS (Advanced TIROS Operational Vertical Sounder) system.

---

## Author Comment (AC3)

This manuscript presents an unprecedented evaluation of 23 (sub)daily (quasi)global precipitation (P)datasets across 16,295 catchments worldwide using hydrological modeling. The 23 P datasets belong to six major families of data sources: satellite only, reanalysis only, rain gauge only, satellite and rain gauge, satellite and reanalysis; and satellite, reanalysis and rain gauge. The conceptual hydrological model HBV was used to simulate the conversion of precipitation into streamflow at the daily temporal scale. Each P dataset, along with air temperature (from MSWX) and potential evapotranspiration (computed using the Hargreaves formula), are used to drive the hydrological simulations. The modified Kling-Gupta efficiency (KGE') is used to evaluate the performance of the simulated streamflows against daily observations, and serves as a proxy for the performance of the P datasets.

This manuscript addresses an important topic for the hydrometeorological community. The manuscript is well written, concise and clear, with updated references. Unfortunately, the manuscript lacks a clear scientific question or hypothesis to be tested, and the Methodology section does not provide enough scientific detail to fully understand what was done and how, which prevents adequate reproducibility of the results. In addition, some conclusions are speculative and are not supported by the results included in the manuscript. Finally, some references are not used in the text and others contain minor errors. To summarise, the manuscript in its current form does not represent a substantial contribution to the global hydrometeorological community; but all the problems mentioned could be addressed by the authors during the review process. The following lines describe the major and minor problems detected in the manuscript.

**Response:** We are thankful to the reviewer for reviewing the manuscript and providing useful comments. We also welcome the critic of the reviewer and suggestions. Your detailed comments have helped us to revisit our analysis and to find some bugs which significantly improved the manuscript. We have revised the manuscript in accordance with your suggestions. We would also like to mention that we have used a revised and updated streamflow database, which includes a higher number of data sources (increased from 22 to 29), improved temporal coverage, and bug fixes—particularly for streamflow records from Africa (ADHI). The extended temporal coverage has led to an increase in the number of catchments with sufficient data for model calibration. As a result, the total number of catchments for which HBV parameters were optimized against observed streamflow increased from 16,295 to 18,428. We have also removed CMORPH-RAW and included CMORPH-CDR, following your recommendation. While the overall findings remain consistent with those in the previous version, we have updated the manuscript wherever new insights emerged. Please note that the line numbers and figure numbers in our responses correspond to the updated manuscript.

Major comments:

1. MC1. The motivation for the article is not well developed. The manuscript does a really good job of pointing out the limitations of previous evaluations of

P datasets. However, what is the ultimate purpose of this comprehensive evaluation of P datasets on a global scale? Is it just to provide some numbers on a global scale, or is it to test a hypothesis or answer a scientific question, or to provide recommendations for the selection of P products for specific applications or specific geographic regions? If so, the hypothesis, the scientific question or the ultimate purpose of the manuscript should be explicitly stated.

**Response:** Thank you for your critical comment. The purpose of this study was to analyze strengths and weaknesses of different P datasets at various geographical and climatological zones and to provide guidance to users on their suitability for various hydrological applications. We have added this sentence in the last paragraph of the introduction section.

Lines 58-60: *"In this study, we present the most comprehensive evaluation to date of gridded (sub-)daily (quasi-)global P datasets, aiming to identify their strengths and limitations across diverse geographical and climatological settings, and to inform their suitability for hydrological applications. "*.

2. MC2. Usage of the outdated CMORPH-RAW (Joyce et al., 2004) and the unknown CMORPH-RT (Xie et al., 2017) instead of the new bias-corrected CMORPH-CDR v.1 (Xie et al., 2017, 2018). In the manuscript it is mentioned that the old CMORPH-RAW and CMORPH-RT are available from 2019 onwards (which seriously limit the hydrological modeling runs), while the newest version of CMORPH, termed CMORPH-CDR, is available from 1998 onwards (not from 2019 onwards). Moreover, it is not clear what is the product CMORPH-RT used in this study, every time that Xie et al. (2017) describe CMORPH-CDR version 1, which is available since 1998 and not from 2019. Therefore, I request the authors to remove the usage of the outdated CMORPH-RAW (version 0) and the unknown CMORPH-RT and use the relatively new bias-corrected CMORPH-CDR version 1, which is available since 1998, and it is described by Xie et al. (2017) and Xie et al. (2018).
   Xie, P., Joyce, R., Wu, S., Yoo, S.-H., Yarosh, Y., Sun, F., and Lin, R.: Reprocessed, Bias-Corrected CMORPH Global High-Resolution Precipitation Estimates from 1998, Journal of Hydrometeorology, 18, 1617–1641, doi:10.1175/JHM-D-16-0168.1, 2017.
   Xie, P., Joyce, R., Wu, S., Yoo, S., Yarosh, Y., Sun, F., Lin, R., and NOAA CDR Program: NOAA Climate Data Record (CDR) of CPC Morphing Technique (CMORPH) High Resolution Global Precipitation Estimates, Version 1, doi:10.25921/W9VA-Q159, URL https://www.ncei.noaa.gov/access/metadata/landing-page/bin/iso?id=gov.noaa.ncdc:C00948, 2018.

**Response:** Thank you for your suggestion. We have included the CMORPH-CDR dataset which starts from 1998 and have removed CMORPH-RAW. However, we have

retained CMORPH-RT. The reason for this being that CMORPH-RT (median KGE: 0.57) performed better than CMORPH-CDR (median KGE: 0.53). The CMORPH-RT dataset refers to satellite only realtime production of CMORPH and was obtained from https://ftp.cpc.ncep.noaa.gov/precip/CMORPH_RT/GLOBE/data/: last accessed July 1 2025. (Joyce et al., 2017). This dataset has recently been used in several studies for evaluation and comparison with other P datasets (Hussain et al., 2018; Liu et al., 2022; Nguyen et al., 2025; Omay et al., 2025). Although the data for CMORPH-RT starts from 2019, the total number of catchments that fulfill the filter criteria for calibration for CMORPH-RT are 7876.

1. MC3. Use of the under-revision PERSIANN-CCS-CDR (Sadeghi et al., 2021). This paper uses PERSIANN-CCS-CDR (Sadeghi et al., 2021) as one of the 23 P datasets to be evaluated. However, the websitehttps://chrsdata.eng.uci.edu/ clearly states that "*PERSIANN-CCS-CDR is currently under revision and unavailable for download*". Therefore, I request the authors to remove the use of PERSIANN-CCS-CDR from this study or clarify the data version used in this study and indicate whether the chosen version is problematic or not.

**Response:** The reason for this revision of PERSIANN-CCS-CDR is that it uses two sources of infrared for different time ranges. For the period from 1983 to February 2000, it uses GridSat-B1 data which is 3-hourly. From March 2000 onwards, it uses NOAA Climate Prediction Center (CPC-4km) data which has 30 minute resolution. This has led to some inconsistencies as reported by Sadeghi et al. (2021). However, the authors of the dataset maintain that the overall performance of their product is consistent at global scale. A new version is expected to overcome these consistencies however it will not be significantly different from the existing version of the product. We have added the clarification in the manuscript

Lines 216 - 218: "*PERSIANN-CCS-CDR is currently under revision due to inconsistencies in the infrared input data before and after 2000 (Sadeghi et al., 2021); however, this issue is unlikely to significantly affect hydrological modeling performance.*"

2. MC4. Catchment selection. To ensure the suitability of the catchments used in the analyses, five selection criteria were applied in the manuscript to the 34,768 streamflow stations that passed the duplication check. However, the following two decisions are entirely subjective and require more detailed explanation (in the manuscript) by the authors: i) discarding streamflow stations where both the station location and the corresponding catchment centroid were within 5 km of those of another station (how does the spatial resolution of the individual P products influence this criterion?); ii) the number of events had to be greater than 10 non-consecutive (how does the duration of each selected event affect this criterion?; are 11 non-consecutive days with

Q >= 5 mm d-1 sufficient to ensure a robust calibration of a hydrological model?)

**Response:** We appreciate the comment. Some of the streamflow sources that we employed for this study contained overlapping stations. For example the GRDC dataset contains catchments from around the world which overlaps with other regional data sources. In order to avoid using the same catchment twice, we performed the duplication check using the method mentioned in the manuscript i.e. if the centroid and outlet of one catchment is within 5 km of another catchment, one of them was discarded. The spatial resolution of P datasets does not affect this filter criteria.

3.  MC5. Use of an unknown version of the HBV hydrological model. The manuscript does not contain a description about the version of the HBV hydrological model used in all analyses. L107 indicates that the HBV-light software described by Seibert and Vis (2012) was the software version used in this study. However, it seems unlikely that a Windows-based version of HBV was selected to simulate 16,295 catchments worldwide. I request the authors to provide details of the version of HBV used in this study. In the event that the authors use their own version of HBV, I request them to provide a link to the source code of the model in the "Code Availability" section requested by HESS (https://www.hydrology-and-earth-system-sciences.net/submission.html#templates).

**Response:** We used a Python version of HBV-light which has been used in several previous studies (Beck et al., 2020; Feng et a., 2024, Feng et al., 2023). For transparency in results we have open sourced the code and it is available at https://github.com/hyex-research/hydro_clim_scen_analysis/blob/main/hbv.py .

4.  MC6. Use of catchment-mean P time series to drive the hydrological model (L89-90, L114-115). The use of catchment mean P time series to drive the hydrological model HBV could lead to important problems in the representation of observed streamflows in catchments with mixed hydrological regimes (i.e. snow-dominated or snow-influenced hydrological regimes), which should be reflected in low KGE values. Therefore, I request the authors to provide -in the supplementary material- five to seven example catchments where the HBV is able to reproduce their mixed hydrological regime by using catchment-mean P time series to drive the hydrological model (I request not only the presentation of the KGE values and the daily time series of the observed Q compared to the simulated Q, but also a comparison of the mean monthly streamflows). If the model is not able to acceptably reproduce the daily and mean monthly observed streamflows of catchments with mixed hydrological regimes, I suggest the authors to implement different elevation bands in these catchments. A publicly available open-source version of an HBV-like hydrological model can be found at: https://cran.rproject.org/package=TUWmodel, which allows the use of up to 10 elevation bands in each catchment.

**Response:** We agree with the reviewer that using catchment averaged forcing data do not account for variations within a catchment such as elevation or land cover change. Implementing a semi-distributed HBV with elevation bands can result in a more representative hydrological model. However, for such a large sample study, calibrating each catchment with elevation bands will significantly increase the computation demand. We therefore consider it to be beyond the scope of this study. However, we agree with the reviewer's suggestion and we have described how the current study can be improved by making use of elevation bands for a future project in section 3.3.

Lines 344 – 348: "*Additionally, it does not account for spatio-temporal variations in land cover or use and relies on catchment-averaged meteorological forcings, omitting sub-catchment variability in climate and terrain. More complex (semi-)distributed models with hydrologic response units or elevation bands may yield improved simulations (Gu et al., 2023). However, we do not expect this to materially affect the relative performance ranking of the P datasets or the main conclusions.*"

5. MC7. Using the Hargreaves (1994) equation to calculate potential evapotranspiration (PET) to drive the hydrological model. I request the authors to justify this choice after knowing that Oudin 2005a proposed a different temperature-based PET model after evaluating 27 potential evapotranspiration models in terms of streamflow simulation efficiency in a large sample of 308 catchments in France, Australia and the United States.

**Response:** We have rerun all the simulations with Penman-Monteith formulation of potential evapotranspiration (PET). Penman-Monteith is a complex method and it requires temperature, relative humidity, radiation and wind speed for calculation of PET. A major conclusion from the study of Oudin et al. (2005) was that the PET methods relying only on radiation and temperature are as efficient as more complex methods. All the figures and tables in our revised manuscript are recreated using the Penman-Monteith method. However, this did not alter the major conclusions of this study or the ranking of P products for different geographical and climatological zones.

6. MC8. Range used for the calibration of the PCORR parameter of HBV. Table 2 shows that the PCORR parameter is used as a multiplier to mitigate the systematic underestimation of P characteristics of some P products, and therefore a range of [1, 2] is used in the optimisation of this parameter. This decision could lead to low KGE values in arid or hyper-arid catchments (see Table 3), where some P datasets overestimate the true (and unknown) P amount. Therefore, I request the authors to extend this range to [0.5, 2] so that the calibration procedure can compensate not only for an underestimation of P but also for an overestimation of it.

**Response:** Thank you for the useful comment. We compared the performance of the P datasets by calibrating KGE values under four different scenarios. In the first scenario, PCORR was allowed to vary between 0.0 and 2.0. In the second scenario, PCORR was restricted to the range 0.5–2.0. In the third scenario, both PCORR and SFCF were fixed at 1.0, as suggested by the reviewer (the 3rd reviewer). The fourth scenario corresponds to the one used in our original setup, where both PCORR and SFCF ranged from 1.0 to 2.0. The resulting ranking of precipitation datasets based on KGE is illustrated in the figure below

[Figure]

**Fig. S26.** Kling-Gupta Efficiency values for P datasets in different scenarios of PCORR and SFCF calibration.

These results indicate that the overall ranking of P datasets do not vary significantly in all four scenarios. However, in scenario three (PCORR: 1.0, SFCF 1.0), the ranking of GPCP V3.2 and ERA5 improved. However, this improvement in ranking of these P datasets is not due to their improved KGE value since the overall median KGE value in scenario three is lower as compared to other three scenarios. The following figure illustrates the decrease in KGE when PCORR and SFCF were kept constant during calibration (third scenario) as compared to the default scenario when their values were allowed to vary between 1.0 to 2.0.

[Figure]

**Fig.** Decrease in KGE values when PCORR and SFCF were kept constant during calibration (1.0) as compared to when they were allowed to vary between 1.0 to 1.0.

The distribution of PCORR values across the four calibration scenarios (as shown in the following figure) reveals that only a few precipitation datasets exhibit significant adjustment below 1.0 when PCORR is allowed to vary from 0.0 to 2.0. Specifically, PDIR-Now, GSMaP V7, PERSIANN-CCS-CDR, and IMERG-Late V6 show a notable portion of calibrated PCORR values falling below 1.0. This indicates that, for these datasets, the raw precipitation input tends to overestimate actual precipitation in a substantial number of catchments.

[Figure]

**Fig S27.** Comparison of distribution of calibrated PCORR values from four calibration scenarios.

We have added the following sentences in our manuscript to describe this.

Lines 138 - 145 : *"To assess the influence of systematic P bias correction using the PCORR and SFCF adjustment factors on model performance, we explored four calibration scenarios with varying bounds for the PCORR and SFCF parameters. In the first scenario, PCORR was allowed to vary between 0.0 and 2.0, providing full flexibility to adjust for both under- and overestimation of P, while SFCF was allowed to vary between 1.0 to 2.0. The second scenario limited PCORR to the range 0.5–2.0, while keeping the range of SFCF between 1.0 and 2.0. The third scenario fixed both PCORR and SFCF parameters at 1.0, effectively disabling P bias correction. The fourth scenario constrained both PCORR and SFCF to the range 1.0–2.0, allowing only upward correction. These scenarios enabled us to evaluate the sensitivity of model performance to P bias correction and assess the robustness of P dataset rankings under varying calibration constraints."*

Lines 240 - 246 : *"The overall ranking of P datasets remained largely consistent across the four PCORR calibration scenarios (Supplement Fig. S26). However, in the scenario where PCORR and SFCF were fixed at 1.0, GPCP V3.2 and ERA5 showed improved relative rankings—not due to higher performance, but because other*

*datasets experienced greater performance drops under this constraint. Most datasets showed little sensitivity to the PCORR bound below 1.0, but a few—namely PDIR-Now, GSMaP V7, PERSIANN-CCS-CDR, and IMERG-Late V6—exhibited notable use of PCORR values below 1.0 (Supplement Fig. S27). This suggests that these datasets tend to overestimate P in certain catchments, and that downward rescaling improves their hydrological performance".*

7.  MC9. Use of an unknown version of the (μ+λ) evolutionary algorithm used to calibrate the HBV hydrological model. The manuscript does not contain a description of the version of the (μ+λ) evolutionary algorithm used to calibrate the HBV hydrological model. From L124, the reader can infer that the DEAP Python software was used to calibrate the HBV model. However, I request the authors to clarify the name and version of the software used to implement the (μ+λ) evolutionary algorithm and to describe how this algorithm was coupled to the (unknown) version of the HBV hydrological model (see MC5). Finally, I request the authors to describe whether they can ensure that the (μ+λ) evolutionary algorithm has converged to a stable KGE value after 1200 model runs (L125) or not.

**Response:** We used version 1.4 of the Distributed Evolutionary Algorithms in Python (DEAP) library. For the HBV, as responded previously, we used the HBV-light version of Seibert and Vis implemented in Python and used in previous studies (Beck et al., 2016, Beck et al., 2020, Feng et al., 2023, Feng et al., 204). During calibration of each catchment, the HBV model was run to simulate streamflow using the parameters suggested by the DEAP algorithm. The simulated streamflow is then compared against observed streamflow to calculate KGE values (Eq. 1 in the manuscript) which is then returned to the optimization algorithm as a feedback signal to its suggested parameters. The evolutionary algorithm of DEAP iteratively improves the suggested parameters which can improve KGE values. In order to ensure the convergence during calibration, we slightly modified the stopping criteria. The minimum number of generations was set to 25 and the calibration process was stopped if the KGE value did not improve more than 0.01 for more than 5 generations. However, to avoid computation loss, we set the maximum number of generations to 40. We found that less than 1% of stations were run for 40 generations which was the upper limit of the number of generations. This showed that for the overwhelming majority of catchments, the convergence was achieved before reaching the maximum number of generations and that further improvements were not possible, indicating that the optimal parameter set has been obtained.

[Figure]

**Fig.** Percentage of stations for which the total number of generations reached 40 during calibration for each P dataset. We set 40 as the maximum number of generations to run for each catchment during calibration.

We have modified the corresponding lines in section 2.4 to elaborate this.

Lines 130 - 137 : "*We used a (μ + λ) evolutionary algorithm, which is a population-based optimization method that iteratively evolves solutions through selection, crossover, and mutation to maximize the Kling-Gupta Efficiency (KGE) objective. The algorithm was implemented using the Distributed Evolutionary Algorithms in Python (DEAP) library (version 1.4; Ashlock, 2010; Fortin et al., 2012), with a population size (μ) of 20 and an offspring pool size (λ) of 48. Crossover was applied with a probability of 90%, and mutation was applied with a probability of 10% using a Gaussian-based mutation operator. To ensure convergence, the optimization process was terminated*

*if the best KGE value did not improve by more than 0.01 for five consecutive generations after a minimum of 25 generations.*"

8. **MC10**. Selection of temporal period used for the calibration of the individual catchments. It is not clear from the manuscript whether the period used to calibrate the HBV hydrological model with each P dataset was the same or whether it depended on the data availability of the respective P product. I request the authors to clarify this situation in the manuscript. In the case that the temporal period used for the calibration of each catchment depends on the data availability of each P product, and therefore, it was not the same for all the P products used as forcing in each catchment, I request the authors to use the same temporal period for the calibration of all P products in each catchment, to ensure a fair comparison of the performance of different P datasets in a given catchment. Of course, the temporal periods may be different from one catchment to another, but for the same catchment the same temporal period should be used to calibrate the HBV model with all P datasets.

**Response:** The temporal range of the data used to calibrate HBV for each P product is different because of differences in the availability period of the P products. Moreover, the temporal range of observed streamflow also varies from catchment to catchment and from data source to data source. Therefore we used the full period of overlapping streamflow and P data for each catchment. Using a common temporal range for all P datasets and all catchments would significantly decrease the number of P datasets and catchments considered in this study, and would result in less generalizable results.

9. MC11. Based on the boxplots summarising the performance of each of the 23 P datasets used in this study, it is quite surprising that the CPC Unified dataset, which is based solely on rain gauge information and has the coarsest spatial resolution of all P datasets (0.5°), ranked second among all datasets. I request the authors to add a paragraph suggesting possible reasons for this unexpected behaviour.

**Response:** You are correct in pointing out that this study demonstrates that datasets with higher spatial resolution do not necessarily result in better performance for hydrological modeling. This finding is consistent with previous studies, including Bador et al. (2020), Huang et al. (2019), and Chan et al. (2013). We have incorporated the following discussion into the manuscript to reflect this point.

*Lines 226 – 232: "Our results reaffirm that higher-resolution P datasets do not necessarily yield better streamflow simulations compared to lower-resolution datasets, consistent with previous assessments (e.g., Bador et al., 2020; Huang et al., 2019; Chan et al., 2013). Notably, the 0.04◦ resolution infrared-based datasets (PERSIANN-CCS and -CCS-CDR, and PDIR-Now;220 median KGE of 0.46, 0.50,*

*and 0.45, respectively) — the highest resolution datasets included in our assessment — do not consistently perform better neither globally nor for any Köppen-Geiger climate zones. This is likely because streamflow aggregates P over space and time, dampening local details captured by high-resolution datasets. Alternatively, coarser datasets may average out small-scale noise, yielding more reliable estimates".*

.

10. MC12. To provide an initial assessment of the ability of all 23 P datasets used in this study to reproduce the mean annual precipitation at a given location, I request the authors to create a new figure with the mean annual precipitation for 2007-2015 (the longest period for which all datasets have data, after removing the two CMORPH products described in MC2), computed as the average of the mean annual values obtained for each of the 23 P datasets for that period (Pavg). In addition, I request the authors to prepare 23 new figures showing the difference between the mean annual precipitation of each P dataset for 2007-2015 (Pi) and Pavg, i.e., Pi - Pavg. All the figures requested in this comment should be included in the supplementary material only, and they will allow to identify major problems in the representation of mean annual values of a given P dataset in some specific regions of the world.

**Response:** We have prepared the figures requested by the reviewer. The mean annual values obtained for each for the 21 P datasets (excluding GDAS and CMORPH-RT) for the period (2007 – 2015) as well as the difference between mean annual P of individual P from the $P_{avg}$ is illustrated in next figures and added in supplementary material.

[Figure]

**Fig S30.** Average of the mean annual precipitation (mm/day) from 23 datasets.

[Figure]

**Fig S31.** Difference between mean annual precipitation of CHIRP and average of mean annual precipitation of all datasets ($P_{avg}$).

[Figure]

**Fig S32.** Difference between mean annual precipitation of CHIRPS V2.0 and average of mean annual precipitation of all datasets ($P_{avg}$).

[Figure]

**Fig S33.** Difference between mean annual precipitation of CMORPH-CDR and average of mean annual precipitation of all datasets ($P_{avg}$).

[Figure]

**Fig S34.** Difference between mean annual precipitation of CPC-Unified and average of mean annual precipitation of all datasets ($P_{avg}$).

[Figure]

**Fig S35.** Difference between mean annual precipitation of ERA5 and average of mean annual precipitation of all datasets ($P_{avg}$).

[Figure]

**Fig S36.** Difference between mean annual precipitation of GPCP V3.2 and average of mean annual precipitation of all datasets ($P_{avg}$).

[Figure]

**Fig S37.** Difference between mean annual precipitation of GPM+SM2RAIN and average of mean annual precipitation of all datasets ($P_{avg}$).

[Figure]

**Fig S38.** Difference between mean annual precipitation of GSMaP V7 and average of mean annual precipitation of all datasets ($P_{avg}$).

[Figure]

**Fig S39.** Difference between mean annual precipitation of GSMaP V8 and average of mean annual precipitation of all datasets ($P_{avg}$).

[Figure]

**Fig S40.** Difference between mean annual precipitation of IMERG-Early V7 and average of mean annual precipitation of all datasets ($P_{avg}$).

[Figure]

**Fig S41.** Difference between mean annual precipitation of IMERG-Final V7 and average of mean annual precipitation of all datasets ($P_{avg}$).

[Figure]

**Fig S42.** Difference between mean annual precipitation of IMERG-Late V6 and average of mean annual precipitation of all datasets ($P_{avg}$).

[Figure]

**Fig S43.** Difference between mean annual precipitation of IMERG-Late V7 and average of mean annual precipitation of all datasets ($P_{avg}$).

[Figure]

**Fig S44.** Difference between mean annual precipitation of JRA-3Q and average of mean annual precipitation of all datasets ($P_{avg}$).

[Figure]

**Fig S45.** Difference between mean annual precipitation of MSWEP V2.8 and average of mean annual precipitation of all datasets ($P_{avg}$).

[Figure]

**Fig S46.** Difference between mean annual precipitation of MSWEP-ng V2.8 and average of mean annual precipitation of all datasets ($P_{avg}$).

[Figure]

**Fig S47.** Difference between mean annual precipitation of PDIR-Now and average of mean annual precipitation of all datasets ($P_{avg}$).

[Figure]

**Fig S48.** Difference between mean annual precipitation of PERSIANN-CCS-CDR and average of mean annual precipitation of all datasets ($P_{avg}$).

[Figure]

**Fig S49.** Difference between mean annual precipitation of PERSIANN-CCS and average of mean annual precipitation of all datasets ($P_{avg}$).

[Figure]

**Fig S50.** Difference between mean annual precipitation of SM2RAIN-ASCAT and average of mean annual precipitation of all datasets ($P_{avg}$).

[Figure]

**Fig S51.** Difference between mean annual precipitation of SM2RAIN-CCI and average of mean annual precipitation of all datasets ($P_{avg}$).

11. MC13. To facilitate the "*generalizability of their findings*" (L50, L57) for readers from different countries, I request the authors to add a new figure to the main body of the manuscript: a map showing, in different colours, the KGE values obtained in each catchment. This figure will allow us to identify the spatial distribution of the high and low performance of each P dataset in the simulation of daily streamflows. This new figure will make it possible to support several statements in the "*Results and Discussion*" section that are currently not supported by any figure in the manuscript.

**Response:** The maps of KGE values along with variability (γ) and bias (β) for each of the 18,428 catchment and for each of the 23 *P* datasets are already illustrated in supplementary material as Fig. S1 – S23.

12. MC14. To facilitate even more the "*generalizability of their findings*" (L50, L57) to readers from the same country but from catchments with different hydrological regimes, I strongly suggest (and do not request) the authors to make an extra effort and classify the hydrological regimes of each of the 16,295 catchments (e.g., pluvial, glacial, snow-dominated, snow-influenced, tropical). This would allow readers to use the results of the articles to select one or more P datasets to use for analysing specific case studies in their own countries. If this suggestion could not be addressed by the authors, I request them to insert three new columns in Table 3: low solid P fraction, medium solid P fraction and high solid P fraction, where the thresholds to distinguish between low, medium and high values of solid P fraction should be proposed by the authors based on their knowledge and the values of solid P fraction of all 16,295 catchments.he values of the solid P fraction of all the 16,295 catchments.

**Response:** In order to analyze the behaviour of P datasets in different climate settings, we divided the catchments according to Köppen-Geiger climate zones i.e. polar, tropical, arid, temperate and continental. The performance of each of the 23 P datasets in these five Köppen-Geiger Zones is illustrated in the following boxplot.

[Figure]

**Fig S23.** Performance of 23 P datasets in major Köppen-Geiger climate zones.

A more detailed spatial KGE map of the best performing P dataset (MSWEP 2.8) in each catchment in each of the Köppen-Geiger zone is shown below. However, even though MSWEP V2.8 performs best, it does not perform best for each of the 18,428 catchments. For this reason we added Fig. 3 in the manuscript which shows the best P dataset for each catchment among the top five P datasets.

[Figure]

**Fig.** Spatial maps and distribution of Kling-Gupta Efficiency for MSWEP V2.8 dataset.

13. MC15a. Poor performance of HBV in arid climates. Although the manuscript does not explicitly mention this, it can be inferred that the authors assume that the performance of HBV is likely to be poor in arid climates (L226), because "*P in arid regions tends to be brief and intense, making it challenging to detect and model accurately(Beck et al., 2017b; Sun et al., 2018; El Kenawy et al., 2019; Beck et al., 2019a)*" (L227-228). However, Seibert and Bergström (2022) mention in their review that the HBV is routinely used to model the impacts of climate change on water resources around the world, including regions as arid as the Nile (Booij et al., 2011) and, threfore, aridity per se should therefore not be a reason to explain a poor performance of the HBV model.

Booij, M. J., Tollenaar, D., van Beek, E., and Kwadijk, J. C. J.: Simulating impacts of climate change on river discharges in the Nile basin, Phys. Chem. Earth, 36, 696–709, https://doi.org/10.1016/j.pce.2011.07.042, 2011.

**Response:** First of all, we would like to mention that we found a bug in our preprocessing of the African streamflow dataset (ADHI). Since the ADHI dataset contributes to most of the stations in arid regions, fixing this bug improved the median KGE for arid regions from 0.51 to 0.60. Second, you are right in pointing out that HBV has been applied around the world. There have been applications of HBV in arid regions but on fewer catchments. There have been very few studies who applied HBV on a global scale. However, the low performance of HBV in arid regions has been reported in several other studies. Feng et al. (2024) applied HBV, LSTM and a hybrid model combining HBV and an LSTM model on a global dataset and found that both perform poorly in arid regions with a median KGE value below 0.4. This indicates that even the data-driven deep learning models such as LSTMs performed poorly in arid regions. Booij et al. (2011) applied HBV in 17 subbasins of the Nile river and obtained a median KGE value of 0.64. However, they had a very small sample size (n=17) as compared to our study (n=1300). By plotting the distribution of KGE for arid regions from all P datasets as boxplot, we can show that KGE for a significant number of catchments went beyond 0.64 as shown in the figure below.

[Figure]

**Fig.** Distribution of KGE values of P datasets in arid regions.

1. MC15b. Definition of the aridity index. In the main text of the manuscript, arid regions are associated with values of the aridity index greater than 1 (L250-251, L266). However, this association is inconsistent with the definition of the aridity index in Table B1 of Appendix B, where the aridity index is defined as the ratio between mean annual P and potential evapotranspiration, and therefore values greater than 1 would indicate wet rather than dry catchments. Please clarify this discrepancy.

**Response:** Thank you very much for highlighting this issue. We have corrected the definition of aridity index in Table B1 now as the ratio of mean PET/P.

2. MC16. Efficiency of the filter used to select the study's catchmens. In Section 3.2 (Regional performance differences) the authors mention aridity, groundwater use and/or anthropogenic water use as possible explanations for the low performance obtained for several P products in Australia, India, South Korea and Africa. Does this mean that the five criteria used in Section 2.2 to "*ensure the suitability of the catchments for the present analysis*" (L87) did not work as expected?. I request the authors to add a discussion of why the five criteria previously mentioned were not sufficient to filter out catchments that were not suitable for the present analysis. I also request the authors to consider whether it is necessary to add one or more criteria that would allow the presence of irrigation, hydrograph regulation and/or major consumptive water use to be detected, in order to screen out catchments that will not provide reliable results from the analysis. I suggest the authors analyse the criteria used by the Reference Observatory of Basins for INternational hydrological climate change detection (ROBIN; Kumar et al., 2024) to ensure that the streamflows observed in each selected catchment are free from anthropogenic influences.

   Kumar, A., Hannaford, J., Turner, S., Barker, L. J., Dixon, H., Griffin, A., Suman, G., and Armitage, R.: Global trend and drought analysis of near-natural river flows: The ROBIN Initiative, EGU General Assembly 2024, Vienna, Austria, 14–19 Apr 2024, EGU24-17249, https://doi.org/10.5194/egusphere-egu24-17249, 2024.

**Response:** The five criteria used in Section 2.2 to ensure the suitability of the catchments for the present analysis (L87) did work according to our expectations. However, it is difficult to discard catchments with small dams, as these small dams are not represented by global dam datasets like GranD (Lehner et al., 2011; as mentioned on lines 304-305). Furthermore, despite the lower performance in arid regions, we did not want to exclude all arid catchments, because arid regions are still hydrologically important, and the performance is not too low (that is, the models still provide useful estimates). Similarly, we did not exclude catchment with considerable irrigation, as this would exclude numerous arid catchments. Regarding water use, there is unfortunately not a sufficiently reliable dataset we can use to account for this, and we do not know what degree of water use would render a catchment meaningless for our analysis.

We also thank you for highlighting the ROBIN project. The stated aim of the ROBIN project is to investigate the impact of climate variability on hydrology. To this end, the project defined two tiers of catchment selection criteria: Tier 1 catchments are intended for the analysis of extreme events, while Tier 2 catchments are used for studying less sensitive hydrological variables. Some of the quantitative selection criteria in the ROBIN project overlap with those used in our study—for example, record length and limits on missing data. Notably, the ROBIN project restricts catchments to those with

urbanization fractions below 10% for Tier 1 and 20% for Tier 2. Although we did not explicitly use urbanization fraction as a selection criterion, only 570 and 206 of our 18,428 selected catchments exceed the 0.1 and 0.2 urbanization thresholds, respectively. Furthermore, we found that excluding these 206 catchments does not significantly affect our overall findings: MSWEP V2.8 remains the top-performing dataset, with CPC Unified, IMERG-Final V7, GDAS, and MSWEP-ng V2.8 consistently ranking among the top five. The performance of all P datasets after excluding catchments with urban fractions greater than 0.2 are shown in the following figure.

[Figure]

Fig. Calibration KGE for catchments with urban fraction less than 0.2.

1. MC17. Make the observed streamflow dataset publicly accessible. HESS request the authors to follow their data policy (https://www.hydrology-and-earth-system-sciences.net/submission.html), which includes a statement on how the underlying research data can be accessed. If the data are not publicly accessible, a detailed explanation of why this is the case is required (e.g. applicable laws, university and research institution policies, funder terms, privacy, intellectual property and licensing agreements, and the ethical context of the research). In addition, the HESS data policy states the provision of unrestricted access to all data and materials underlying reported findings for which ethical or legal constraints do not apply. It is true that a URL or reference to the data source of the streamflow data used in this study is provided in Table A1. However, a researcher wishing to reproduce the results

of this study will never be certain that the data downloaded from each URL corresponds exactly to the original 43,627 stations used in this study. Furthermore, in the hypothetical situation of having downloaded exactly the same 43,627 stations that were originally used in this study, it would not be possible to ensure that applying the five criteria, presented in Section 2.2 for filtering out stations, would result in exactly the 16,295 stations finally analysed in this study. Therefore, in practise, it would not be possible for a researcher to reproduce the results of this study. The entire scientific community will thank the authors of this study for providing public access to the daily streamflow data, the catchment boundaries and the location of the outlet of each catchment in order to improve this dataset for future analyses on a global scale.

**Response:** We agree that open access to such a large corpus of streamflow data would greatly benefit the scientific community. However, the authors do not have the necessary permissions to share the observed streamflow data used in this study. The data were either obtained from the meteorological websites of respective countries or sourced from publicly available datasets such as CAMELS. The Python code used to fetch and harmonize the streamflow data from these sources is openly available in the AquaFetch GitHub repository (Abbas et al., 2025; https://github.com/hyex-research/AquaFetch). While the exact number of stations may vary depending on the length of available records, this variability does not significantly affect the findings of the study, as stations with short records were already excluded from the analysis.

Minor comments:

1. In all the manuscript, I ask the authors to use the word "*reanalysis*" instead of "*model*" when referring to atmospheric models of the global climate (e.g., ERA5, JRA-3Q), to avoid confusion with the HBV hydrological model used in this study.

**Response:** We have replaced the word "*model*" with reanalysis or analysis in the whole manuscript. The word "*model*" is now solely reserved for HBV.

2. Provide the full name of all the abbreviations used in the manuscript the first instance they appear, as specified in the "*English guidelines and house standards*" of HESS (https://www.hydrology-and-earth-system-sciences.net/submission.html). This is particularly important for all the precipitation products, which can not be assumed to be known by the wider scientific community. In addition, please provide a reference for each P dataset the first time they appear in the text.

**Response:** Thank you for pointing this out. We have provided the full names of all dataset abbreviations in the abstract. Additionally, we have included a separate column in Table 2 that lists the full name of each dataset along with its corresponding reference.

3. Because CAMELS is a catchment dataset specifically developed for U.S., I request the authors to use CAMELS only for the US datasets, while when referring to CAMELS-like datasets developed for other countries, the individual names of the datasets should be used (e.g., CAMELS-GB, CAMELS-CL) or a generic name different from "CAMELS".

**Response:** Thank you for pointing out that the original CAMELS dataset was developed for the U.S. Since then, several CAMELS variants have been created for other regions. Based on the reviewer's suggestion, we have adopted more specific names for each CAMELS-like dataset, such as CAMELS-CL for Chile and CAMELS-GB for Britain. Please note that we did not use the CAMELS-US dataset (Addor et al., 2015) due to its limited coverage (561 stations, ending in 2014). Instead, we obtained streamflow data for U.S. stations directly from the USGS website, and the corresponding catchment boundaries were acquired from the HYSETS dataset (Arsenault et al., 2020).

4. To avoid possible ambiguities, use always in the text "*streamflow*" instead of "*flow*". Also, when using "*runoff*" instead of "*streamflow*", specify how runoff was obtained.

**Response:** Thank you for pointing this out. We have replaced the word "*flow*" with "*streamflow*" in the whole manuscript. The runoff was calculated using streamflow record and area of catchment. We have elaborated it in Table B1 of the manuscript.

5. L20-21. Provide a reference for the crucial role that the spatio-temporal distribution of P plays in water resources assessment.

**Response:** We have added two more references in this sentence. The study of Dresel et al., (2018) investigated the effect of precipitation variability on water yield of multiple catchments in Victoria, Australia. McKinnon and Deser, 2021 studied the implications of precipitation distribution for water resources of Western U.S.

Lines 20 – 22: *"Understanding the spatio-temporal distribution of precipitation (P) is crucial for a wide range of applications, including water resources assessment, flood forecasting, agricultural monitoring, and disease tracking (Dresel et al., 2018; Liang and Gornish, 2019; McKinnon and Deser, 2021; Hinge et al., 2022; Dimitrova et al., 2022)".*

6. Table 1. Correct the reference provided for IMERG-Final V7, because Huffman et al. (2019) makes reference to version 6 and not to version 7.

**Response** : As per the release notes of IMERG V7 (https://gpm.nasa.gov/resources/documents/imerg-v07-release-notes: Last accessed 30 April 2025), the core algorithm concept of IMERG is agnostic to version changes.

7. Table 1. In the column "Temp. Cov.", please explain the meaning of "NRT" in the caption of the table, and remove that term (assumed to mean "near real-time") for all the products which time latency is larger than 1 day.

**Response** : We have added the meaning of NRT in the caption of Table 1 and also removed NRT from P datasets with latency larger than 1 day.

8. Table 1. Provide a "Time Latency" value for all the products lacking such information.

**Response** : Table 1 has been modified to include latency information of all products which are still updated.

9. Table 1, Table 3, Figure 2. Please check whether IMERG-Early V7 was used in this study or not, because L72 mentions only IMERG-Early V6 and not IMERG-Early V7.

**Response** : Thanks for highlighting this issue. We used IMERG-Early V7 and not IMERG-Early V6. We have replaced IMERG-Early V6 with IMERG-Early V7 in L72.

10. L69. It mentions that "*The datasets fall into two main categories*". However, in L149 it is mentioned that "*Among the six main categories of P datasets*", which is consistent with the six categories used in Table 1 (column '*Data Source*') and Table 3 (column '*Dataset Type*'). I ask the authors to keep six categories in all the manuscript, using '*Dataset Type*' as a consistent denomination name and using "S, R (reanalysis), G, S+G, S+R, S+R+G" as possible values for this denomination name (instead of "S; R (reanalysis); G; S,G; S,R; S,R,G" as used in Table 1).
CAMELS-like instead of CAMELS.

**Response** : We have modified the sentence as recommended by the reviewer.

Lines 75 – 76: *"The 23 P datasets are grouped into six categories based on their input data sources (see Table 2 for full dataset names and references".*

11. L82. Change "and websites" by "or websites", because Table A1 provide either a reference or a URL but not both.

**Response** : We have modified the sentence.

"*Appendix A provides a detailed list of the data sources, along with corresponding references or websites*".

12. L103-104. Provide the catchments areas corresponding to the 2.5 and 97.5 percentiles as well.

**Response :** The area corresponding to 2.5 and 97.5 percentiles 23 and 6165 $Km^2$. We have revised the sentence to include these two values.

13. Figure 1. Explain in the caption what is specifically shown in panels a) and b) of this figure.

**Response:** Fig. 1a indicates locations of all 34, 768 gauge stations while Fig. 1b indicates the dominant Köppen-Geiger climate class based on the 1-km resolution map from Beck et al., (2023).

14. Table 2. Please add a new column "units" to specify the measurement units of each HBV parameter.

**Response:** Although we had mentioned the units of the HBV parameters in parenthesis, we have now added a separate column for the units based upon reviewer's suggestion.

15. L122-124. Provide more details about the statement: "*Model initialization was done by running the model with 10 years of prior P data, if available. If 10 years of prior P data were not available, the model was run multiple times using the available P data until a total of more than 10 years was accumulated*". In particular, clarify how running multiple times the HBV model allow to compensate the lack of P data.

**Response:** The warm-up period is an adjustment process for the model storages to reach from an empty to an "equilibrium" state. A typical warm-up period ranges from one to several years however it leads to important data loss. Running the model repeatedly will eventually lead the internal storages of the model to their optimal state.

16. L130. Remove GDAS from the examples of P datasets with short record, because its data start in 2001, in contrast to the two CMORPH versions which data starts in 2019.

**Response:** In this study we used GDAS which is based on the most recent V16.3 from 2022 of the Global Forecasting System (GFS) Numerical Weather Prediction (NWP) model. We did not use the data from 2001 which is based on a very old model and that model is not representative of the current model. We have corrected that information in Table 1.

17. L144. Explain what do you mean by "*γ reflects the shape of P probability distribution*".

**Response:** We have modified this confusing part of part of the sentence as below:

Lines 162 – 165: "*While the PCORR and SFCF parameters, which account for systematic biases, were calibrated, the β component of KGE reflects residual biases that may persist due to limitations in the P dataset's ability to accurately represent the spatial and temporal distribution of precipitation intensities and magnitudes (Sun et al., 2017)*".

18. L158. In the sentence "*Specifically, gauge data enhance performance in …*" do you mean something like "*Specifically, bias correction using gauge data enhance performance in ….*"?

**Response:** Yes you are right. The gauge-based P datasets can benefit from bias-correction from gauge-dense regions. We have improved the sentence in the manuscript to avoid this ambiguity.

Line 180: "*Specifically, bias correction using gauge data enhances performance in regions with dense rain gauge networks...* "

19. L165-166. Please provide a reference that support the statement about the climatological rain gauge adjustment in IMERG-Late V7. This is requested because to the best of my knowledge the document "IMERG_V07_ReleaseNotes_final_230713.pdf", only mentions "*Applied climatological adjustment to the Final Run for Early and Late Runs*".

**Response:** Thank you for your concern. This is mentioned in section 3.9 of Huffman et al., (2023). We have added this reference to support this statement.

20. L174-175. Provide a discussion about the poor performance of PDIR-now in UK, Denmark and Italy.

**Response:** A detailed analysis of PDIR-Now's performance in the UK, Denmark, and Italy indicates that its lower performance can be attributed to the restricted lower bound of PCORR during calibration. Initially, the lower limit of PCORR was set to 1.0; however, when this limit was reduced to 0.0, the calibrated PCORR values decreased. The median calibrated PCORR values dropped to 0.76 for CAMELS-GB, 0.74 for Denmark, and 1.04 for Italy. As a result, the median KGE values for these countries improved to 0.39, 0.49, and 0.37, respectively. These findings suggest a consistent overestimation of precipitation by PDIR-Now in these regions.

[Figure]

**Fig S29.** Comparison of calibrated PCORR and corresponding KGE values from 531 catchments from CAMELS-GB using PDIR-Now P dataset.

We have added the following sentence in section 3.1

Lines 250 – 251 : "*Further analysis revealed that the largest decrease in median calibrated PCORR (1.0 to 0.7) and consequently improvement in KGE (0.15 to 0.37) was observed in CAMELS-GB (Supplement Fig. S29).*"

    21. L179. Provide a reference for GDAS.

**Response:** There is no peer-reviewed scientific publication introducing the GDAS precipitation dataset. The dataset is provided by the National Centers for Environmental Information (NCEI) of NOAA, United States. We have added the corresponding website reference in the manuscript.

22. L202. Could you be more specific with the sentence "*the importance of improving coverage in data sparse regions due to data sharing limitations*" ?

**Response:** Thank you for the suggestion. We have revised the sentence to clarify that the accuracy of gauge-corrected precipitation datasets depends heavily on the density and availability of ground-based rain gauge data. In many regions—particularly in parts of Africa, South America, and central Asia—limited data sharing policies or inadequate observational infrastructure restrict the integration of gauge data into global products. This leads to reduced correction quality and poorer performance in these regions. Therefore, expanding both the spatial coverage and accessibility of gauge observations is essential for improving the performance of precipitation datasets globally.

Lines 212 – 225 : "*This highlights the critical role national meteorological agencies play in feeding rain gauge data into global databases such as the Global Historical Climatology Network daily (GHCNd; Menne et al., 2012) and the need to expand gauge coverage and promote open data sharing, particularly in data-scarce regions, to improve the accuracy of P datasets in those areas.*"

23. L203. Where can we see the "*comparison of PCORR parameter values obtained after calibration using different P datasets*" ?

**Response:** The spatial distribution of PCORR is shown in supplementary figures Figs S1-S23. However, for ease of comparison, we have also added the boxplot comparing the distribution of PCORR values obtained from all P datasets in supplementary.

24. L209. How is it possible to obtain negative values of the PCORR parameter if the range specified for this parameter in Table 2 was [1, 2]?

**Response:** We apologize for the confusion. The sentence in line 209 does not refer to negative PCORR values but to lower bias (beta) values for IMERG Early and Late V7 products (Fig. 2c). We have modified the sentence to correct this.

25. Figure 2. Add to the caption of this figure the meaning of the horizontal black line shown in each boxplot.

**Response:** The horizontal black line in each boxplot indicates the mean value. We have updated the caption to include this information.

26. L237-L241. To avoid confusion, please use the same attribute names used in Figure 4 and Appendix B (e.g., use "*Mean PET*" instead of "*low Mean PET*").

**Response:** The word "*low* " in the term "*low Mean PET*" here referred to lower values of mean potential evaporation values (*Mean PET)*. Please note that we have replaced the word evapotranspiration with evaporation following the recommendations of Miralles et al., (2020). Both Fig. 4 and Table in appendix B has been updated.

27. L240. Develop more the idea "…, *as frontal P is prevalent under these conditions*".

**Response:** The high solid precipitation fraction and low mean temperature are associated with frontal precipitation which affects ERA5 performance. Frontal precipitation typically occurs in mid- to high-latitude regions, where colder temperatures and snow-dominated conditions prevail (Milani and Kidd, 2023; Hénin et al., 2019). These large-scale frontal systems are well captured by reanalysis models like ERA5, leading to better agreement with observed streamflow.

28. L243. Please introduce the concept "*Rain Gauge Density map*" before using it here.

**Response** : We have modified the sentence as below

Lines 294 – 296: *"Rain Gauge Density, calculated as the number of gauges per 100 km2 smoothed using an exponential filter (see Table B1 for details), showed a slight positive relationship with MSWEP v2.8 performance, suggesting that a higher gauge density contributes to improved accuracy, as expected".*

29. L272. Correct "JRA-3"

**Response** : We have corrected it. The revised sentence is below

*Lines 323 – 325 : "For (re)analysis-based datasets (ERA5, GDAS, and JRA-3Q), limited availability of surface, radiosonde, and aircraft observations for assimilation further reduces performance (https://charts.ecmwf.int/catalogue/packages/monitoring/)".*

30. L275. Explain the meaning of TOVS-to-ATOVS.

**Response**: The TOVS-to-ATOVS transition in ERA5 refers to a change in satellite observation systems. TOVS (TIROS Operational Vertical Sounder) was an older generation of sounders used from the 1970s to the 1990s. In 1998–1999, it was replaced by the more advanced ATOVS (Advanced TIROS Operational Vertical Sounder) system.

31. L277-280. Where can we see the low performance obtained by PDIR-Now in Italy and Denmark, as well as the low performances obtained by JRA-3Q in Tahiland?

**Response** : The lower performance of PDIR-Now in Italy and Denmark as well as the lower performance of JRA-3Q in Thailand is visible in heatmap Fig. 5. This heatmap indicates calibration KGE between streamflow data sources and P products. The dark blue color indicates lower KGE values. The exact median KGE values of PDIR-Now for Italy and Denmark are 0.27 and 0.29 respectively. For JRA-3Q, the median KGE value is -2.6.

32. L288. To improve the clarity of the text, please change "*bias-adjustment techniques*" by "*bias-adjustment techniques of P datasets*".

**Response:** We have modified the sentence as you have recommended.

33. L310-312. Can you provide any number to support the statement "*our approach may slightly overestimate the relative performance of gauge-based and model-based datasets compared to satellite-only datasets*"?

**Response:** To support our argument, we compared the performance of precipitation datasets in regions with differing rain gauge densities. Specifically, we plotted KGE values for the 100 catchments with the highest gauge density and the 100 catchments with the lowest gauge density (see figure below). For the CPC Unified dataset, which is purely gauge-based, its ranking dropped from 2nd place in high-density regions to 7th place in low-density regions. Conversely, the relative performance of satellite-only datasets such as GPM+SM2RAIN and SM2RAIN-CCI improved in areas with sparse rain gauge coverage. This demonstrates that gauge-based datasets tend to perform better in regions with high gauge density, potentially leading to an overestimation of their relative performance when evaluated globally.

[Figure]

**Fig.** Comparison of P datasets for regions with high rain gauge density vs regions with low rain gauge density.

34. L313. Remove GDAS from the examples of P datasets with short record, because its data start in 2001, in contrast to the two CMORPH versions which data starts in 2019.

**Response:** In this study we used GDAS which is based on the most recent V16.3 from 2022 of the Global Forecasting System (GFS) Numerical Weather Prediction (NWP) model. We did not use the data from 2001 which is based on a very old model and that model is not representative of the current model.

35. L317-321. I suggest to move these lines into a new section termed "Future work".

**Response:** Since there are some other limitations of this work as well which – as we have indicated – can be overcome in a future work, we have modified the title of this section to "Potential Limitations and Future Work"

36. L331. Given that GPM+SM2RAIN performed best among all the satellite-only P datasets, and considering that the developers of that product are among the authors of this work, can you provide some description of the reasons that prevent updating this product at least once a year?

**Response:** GPM+SM2RAIN was a prototype of an European Space Agency (ESA) project that for being updated and maintained, would necessitate funds for the repossessing of all the products used for it. However, currently this activity is not funded by ESA and the processing of the product is therefore halted.

37. L334. Stating that MSWEP is a "gauge-based" dataset gives the wrong idea that this product is only based on rain gauge information. I suggest to be more specific here and specify that this product uses information from rain gauges, among other sources.

**Response:** This sentence compares precipitation datasets that incorporate gauge correction. To avoid confusion, we have revised the sentence as follows:

" MSWEP V2.8 led among the datasets which apply gauge correction, benefiting from its daily gauge corrections, unlike others with five-day or monthly gauge correction."

38. L339-340. The statement "*while arid regions exhibited overall poor performance, with model-based datasets slightly outperforming others*" is not correct, because Table 3 shows that IMERG-Final V7, GPCP v3.2 and CPC Unified outperformed reanalysis datasets in arid regions. Please correct.

**Response:** Thank you for highlighting this. Among the five Köppen-Geiger climate zones, the lowest KGE (median KGE: 0.60) was obtained in arid regions (Table 3). We have modified the sentence to remove the later part which is incorrect.

" ... while arid regions exhibited overall poor performance".

Please note that we have replaced the word "model" with "reanalysis" as you recommended in your other comment.

39. In the sections "*Results and Discussion*" and "*Conclusions*" please provide some analysis of the performance of the P datasets in mountainous regions, which is of utmost interest for the wider hydrological community.

**Response:** We analyzed the behaviour of all 23 P datasets with variation in slope and the results are illustrated in the following figure as bar charts. The first part of the figure indicates the difference in HBV performance between catchments with average slope less than 1 degree and the catchments with average slope greater than 10 degrees. The negative values indicate that the performance increased for catchments with

higher slope while positive values indicate decrease in performance in catchments with average slope greater than 10 degrees. The second part of the figure indicates the spearman correlation between HBV performance (median KGE) and slope. These figures indicate that the performance of P datasets which do not involve gauge-correction increases with slope while the performance decreases with slope for catchments which involve gauge correction. We have added this in "Results and Discussion" section of the manuscript.

[Figure]

**Fig S52.** Variation in HBV performance (KGE) with increase in slope.

Lines 254 - 261 : "*The difference in model performance (median KGE) between flat and steep catchments (average slope > 10°) indicates that the performance of gauge-corrected P datasets tends to decrease in steep regions (Supplement Fig. S52a). This is reflected by a positive difference in median KGE between flat and steep catchments for gauge-corrected datasets. In contrast, for non-gauge-corrected P datasets, this difference is negative, indicating an increase in performance in steeper catchments. This pattern is further supported by the negative correlation between model performance (median KGE) and slope for most gauge-corrected datasets (Supplement Fig. S52b). This decrease in performance of gauge-corrected P datasets in mountainous regions can be explained by the sparsity of rain gauges (Kidd et al., 2017), which limits the effectiveness of gauge correction in those areas.*"

40. In the Section "*Conclusions*" please mention something about the catchment attributes that would allow to predict -to some extent- a good performance of

the P datasets, which is of utmost interest for the wider hydrological community.

**Response:** We have added the following sentence in conclusion to describe the best predictors for high KGE of MSWEP V2.8 which is the best performing dataset.

Lines 377 – 378 : *The best predictors for high KGE of MSWEP~V2.8 are high Mean NDVI and Mean LAI as well as low Mean PE and low Aridity Index*.

41. L359. NOAA is written twice. Correct.

**Response** : Thanks for highlighting the typo. We have corrected the sentence by removing duplication of NOAA. The revised sentence is as below

*CPC Unified is available on the NOAA Physical Sciences Laboratory (PSL) website (https://psl.noaa.gov/data/gridded/data.cpc.globalprecip.html .*

42. L373. Change the capital "*O*" used in "*Observed*".

**Response** : Thanks for the highlight. We have corrected the sentence. The corrected sentence is as below

*For the remaining sources, except GRDC, daily observed streamflow data were obtained from the websites of the respective countries' hydrological or meteorological agencies.*

43. L377. Mention in the text where the radiation and humidity data are used in this work.

**Response** : The radiation and humidity data was used to calculate evaporation according to the penman-monteith method (FAO-1998). We have stated this in the text.

44. Table A1. Please separate the "Data source" column into two different columns: "Institution name" and "Country", to have better information about the data source used for the observed streamflow data.

**Response:** Thank you for your suggestion. However, not every entry in Table A1 is suitable for inclusion in the "Institution Name" column. For example, several datasets were obtained from previously published studies involving collaborations among multiple institutions. To enhance clarity, we have instead included a separate column specifying the spatial coverage of each dataset, since it is not evident from the dataset name.

45. Table B1. Indicate the measurement unit used for the attribute "*Rain gauge density*".

**Response:** The units of rain gauge density are the number of gauges per 100 $Km^2$. We have mentioned it in Table B1.

46. Table B1. Incorrect citation to Legates and Bogart (2009). Please correct.
**Response:** Thanks for the highlight. We have corrected the reference.

"*Fraction of total P falling as snow calculated according to Legates and Bogart (2009)*
*using WorldClim V2 (Fick and Hijmans, 2017) for land and ERA5 (Hersbach et al., 2020)*
*for ocean*".

47. Table B1. Considering the existence of the attribute "*Permafrost fraction*", why the attribute "*Glacier fraction*" was not included in the analysis?
**Response:** The reason for not including "glacier fraction" in Table B1 is that its behavior closely mirrors that of "permafrost fraction." To support this point, we calculated the glacier fraction using the Randolph Glacier Inventory dataset Version 7.0 (RGI 7.0, 2023). The relationship between HBV performance (median KGE) and both permafrost and glacier fractions is shown in the following figure. These charts indicate that for datasets such as CMORPH-CDR, MSWEP V2.8, CPC-Unified, and GPM+SM2RAIN, both glacier and permafrost fractions exhibit a negative correlation with KGE. In contrast, the remaining datasets show a positive correlation with both variables. This similarity in behavior suggests that permafrost and glaciers exert comparable influences on model performance, and therefore, we opted to include only the permafrost fraction in our analysis.

[Figure]

**Fig.** Variation in HBV performance (median KGE) with permafrost fraction and glacier fraction for all 23 P datasets.

48. L388-394. Please provide the correct acknowledgment to each one of the P datasets used in this study, as requested by each data source provider.

**Response:** We have expanded the Acknowledgements section to properly recognize each precipitation dataset provider, in accordance with the guidelines of the respective organizations wherever applicable.

49. L399. There is an incorrect character in the reference. Correct it.

**Response:** We have corrected the reference. The updated reference is as below

*Aouissi, J., Benabdallah, S., Lili Chabaâne, Z., and Cudennec, C.: Evaluation of potential evapotranspiration assessment methods for hydrological modelling with SWAT — Application in data-scarce rural Tunisia, Agricultural Water Management, 174, 39–51, https://doi.org/10.1016/j.agwat.2016.03.004, 2016*

50. L503-508. This reference is repeated twice. Correct it.

**Response :** Thank you for the highlight. We have removed the duplication in reference.

51. L612-615. This reference is repeated twice. Correct it.

**Response :** Thank you for the highlight. We have removed the duplication in reference.

52. L631. Correct the error in the URL.

**Response:** We have corrected the URL and removed the repeated https at the start of url. The corre is"https://doi.org/10.1016/j.jhydrol.2021.126455"

**References**

Abbas, A., Iftikhar, S., & Beck, H. E. (2025). AquaFetch: A Unified Python Interface for Water. *Journal of Open Source Software.* Under Review

Addor, N., Newman, A. J., Mizukami, N., & Clark, M. P. (2017). The CAMELS data set: catchment attributes and meteorology for large-sample studies. *Hydrology and Earth System Sciences*, *21*(10), 5293-5313.

Arsenault, R., Brissette, F., Martel, J. L., Troin, M., Lévesque, G., Davidson-Chaput, J., ... & Poulin, A. (2020). A comprehensive, multisource database for hydrometeorological modeling of 14,425 North American watersheds. *Scientific Data*, *7*(1), 243.

Bador, M., Boé, J., Terray, L., Alexander, L. V., Baker, A., Bellucci, A., Haarsma, R., Koenigk, T., Moine, M.-P., Lohmann, K., et al.: Impact
of higher spatial atmospheric resolution on precipitation extremes over land in global climate models, Journal of Geophysical Research:415
Atmospheres, 125, e2019JD032 184, 2020.

Beck, H. E., van Dijk, A. I., De Roo, A., Miralles, D. G., McVicar, T. R., Schellekens, J., & Bruijnzeel, L. A. (2016). Global-scale regionalization of hydrologic model parameters. *Water Resources Research*, *52*(5), 3599-3622.

Beck, H. E., Westra, S., Tan, J., Pappenberger, F., Huffman, G. J., McVicar, T. R., Gründemann, G. J., Vergopolan, N., Fowler, H. J., Lewis, E., Verbist, K., and Wood,

E. F.: PPDIST, global 0.1∘ daily and 3-hourly precipitation probability distribution climatologies for 1979–2018, 425 Scientific Data, 7, https://doi.org/10.1038/s41597-020-00631-x, 2020.

Beck, H. E., McVicar, T. R., Vergopolan, N., Berg, A., Lutsko, N. J., Dufour, A., ... & Miralles, D. G. (2023). High-resolution (1 km) Köppen-Geiger maps for 1901–2099 based on constrained CMIP6 projections. *Scientific data*, *10*(1), 724.

Chan, S. C., Kendon, E. J., Fowler, H. J., Blenkinsop, S., Ferro, C. A., and Stephenson, D. B.: Does increasing the spatial resolution of a regional climate model improve the simulated daily precipitation?, Climate dynamics, 41, 1475–1495, 2013.460

Dresel, P. E., Dean, J. F., Perveen, F., Webb, J. A., Hekmeijer, P., Adelana, S. M., and Daly, E.: Effect of Eucalyptus plantations, geology,530 and precipitation variability on water resources in upland intermittent catchments, Journal of hydrology, 564, 723–739, 2018.

Feng, D., Beck, H., Lawson, K., & Shen, C. (2023). The suitability of differentiable, physics-informed machine learning hydrologic models for ungauged regions and climate change impact assessment. *Hydrology and Earth System Sciences*, *27*(12), 2357-2373.

Feng, D., Beck, H., De Bruijn, J., Sahu, R. K., Satoh, Y., Wada, Y., ... & Shen, C. (2024). Deep dive into hydrologic simulations at global scale: Harnessing the power of deep learning and physics-informed differentiable models (δ HBV-globe1. 0-hydroDL). *Geoscientific Model Development*, *17*(18), 7181-7198.

Gu, L., Yin, J., Wang, S., Chen, J., Qin, H., Yan, X., He, S., and Zhao, T.: How well do the multi-satellite and atmospheric reanalysis products perform in hydrological modelling, Journal of Hydrology, 617, 128 920, 2023.

Hénin, R., Ramos, A. M., Schemm, S., Gouveia, C. M., & Liberato, M. L. (2019). Assigning precipitation to mid-latitudes fronts on sub-daily scales in the North Atlantic and European sector: Climatology and trends. *International Journal of Climatology*, *39*(1), 317-330.

Hussain, Y., Satgé, F., Hussain, M. B., Martinez-Carvajal, H., Bonnet, M. P., Cárdenas-Soto, M., ... & Akhter, G. (2018). Performance of CMORPH, TMPA, and PERSIANN rainfall datasets over plain, mountainous, and glacial regions of Pakistan. *Theoretical and applied climatology*, *131*, 1119-1132.

Huang, Y., Bárdossy, A., and Zhang, K.: Sensitivity of hydrological models to temporal and spatial resolutions of rainfall data, Hydrology and Earth System Sciences, 23, 2647–2663, 2019.550

Huffman, G.J., D.T. Bolvin, R. Joyce, E.J. Nelkin, J. Tan, D. Braithwaite, K. Hsu, O.A. Kelley, P. Nguyen, S. Sorooshian, D.C. Watters, B.J. West, P. Xie, 2023: Algorithm Theoretical Basis Document (ATBD) NASA Global Precipitation Measurement (GPM) Integrated Multi-satellitE Retrievals for GPM (IMERG) Version 07. PPS, last updated 12 July 2023. 52 pp. https://gpm.nasa.gov/resources/documents/imerg-v07-atbd

Joyce, R., Xie, P., & Wu, S. (2017, December). A Preliminary Examination of the Second Generation CMORPH Real-time Production. In *AGU Fall Meeting Abstracts* (Vol. 2017, pp. H31L-05).

Lehner, B., Reidy Liermann, C., Revenga, C., Vörösmarty, C., Fekete, B., Crouzet, P., Döll, P., Endejan, M., Frenken, K., Magome, J., Nilsson, C., Robertson, J. C., Rödel, R., Sindorf, N., and Wisser, D.: High resolution mapping of the world's reservoirs and dams for 580 sustainable river flow management, Frontiers in Ecology and the Environment, 9, 494–502, 2011.

Liu, S., Wang, J., & Wang, H. (2022). Assessing 10 satellite precipitation products in capturing the july 2021 extreme heavy rain in Henan, China. *Journal of Meteorological Research*, *36*(5), 798-808.

McKinnon, K. A. and Deser, C.: The inherent uncertainty of precipitation variability, trends, and extremes due to internal variability, with implications for Western US water resources, Journal of Climate, 34, 9605–9622, 2021.

Menne, M. J., Durre, I., Vose, R. S., Gleason, B. E., and Houston, T. G.: An overview of the Global Historical Climatology Network-Daily database, Journal of Atmospheric and Oceanic Technology, 29, 897–910, 2012a

Milani, L., & Kidd, C. (2023). The state of precipitation measurements at mid-to-high latitudes. *Atmosphere*, *14*(11), 1677.

Miralles, D. G., Brutsaert, W., Dolman, A. J., & Gash, J. H. (2020). On the use of the term "evapotranspiration". *Water Resources Research*, 56(11), e2020WR028055. https://doi.org/10.1029/2020WR028055

Nguyen, V. D., Rouzegari, N., Dao, V., Almutlaq, F., Nguyen, P., & Sorooshian, S. (2025). Comparative Analysis of Satellite-Based Precipitation Products During Extreme Rainfall from Super Typhoon Yagi in Hanoi, Vietnam (September 2024). *Remote Sensing*, *17*(9), 1598.

Omay, P. O., Muthama, N. J., Oludhe, C., Kinama, J. M., Artan, G., & Atheru, Z. (2025). Evaluation of satellite-based rainfall estimates over the IGAD region of Eastern Africa. *Meteorology and Atmospheric Physics*, *137*(2), 1-26.

Oudin, L., Hervieu, F., Michel, C., Perrin, C., Andréassian, V., Anctil, F., & Loumagne, C. (2005). Which potential evapotranspiration input for a lumped rainfall–runoff model?: Part 2—Towards a simple and efficient potential evapotranspiration model for rainfall–runoff modelling. *Journal of hydrology*, *303*(1-4), 290-306.

Pereira, L. S., Perrier, A., Allen, R. G., & Alves, I. (1999). Evapotranspiration: concepts and future trends. *Journal of irrigation and drainage engineering*, *125*(2), 45-51.

RGI 7.0 Consortium, 2023. Randolph Glacier Inventory - A Dataset of Global Glacier Outlines, Version 7.0. Boulder, Colorado USA. NSIDC: National Snow and Ice Data Center. doi:10.5067/f6jmovy5navz. Online access: **https://doi.org/10.5067/f6jmovy5navz**

Seibert, J. and Vis, M. J. P.: Teaching hydrological modeling with a user-friendly catchment-runoff-model software package, Hydrology and Earth System Sciences, 16, 3315–3325, 2012.

Sadeghi, M., Nguyen, P., Naeini, M. R., Hsu, K., Braithwaite, D., and Sorooshian, S.: PERSIANN-CCS-CDR, a 3-hourly 0.04° global precipitation climate data record for heavy precipitation studies, Scientific Data, 8, 157, https://doi.org/10.1038/s41597-021-00940-9, publisher: Nature Publishing Group, 2021.